# Prompt-based Adaptation in Large-scale Vision Models: A Survey

Project Page: https://yunbeizhang.github.io/Awesome-Visual-Prompt-Tuning/

**Xi Xiao**[*]                                                          *xxiao@uab.edu*
*University of Alabama at Birmingham, USA*

**Yunbei Zhang**[*]                                                     *yzhang111@tulane.edu*
*Tulane University, USA*

**Lin Zhao**[*]                                                         *zhao.lin1@northeastern.edu*
*Northeastern University, USA*

**Yiyang Liu**[*]                                                       *yl93b@umkc.edu*
*University of Missouri-Kansas City, USA*

**Xiaoying Liao**                                                       *xliao13@jh.edu*
*Johns Hopkins University, USA*

**Zheda Mai**                                                           *mai.145@osu.edu*
*Ohio State University, USA*

**Xingjian Li**                                                         *lixj04@gmail.com*
*Carnegie Mellon University, USA*

**Xiao Wang**                                                           *wangx2@ornl.gov*
*Oak Ridge National Laboratory, USA*

**Hao Xu**                                                              *haxu@bwh.harvard.edu*
*Harvard University, USA*

**Jihun Hamm**                                                          *jhamm3@tulane.edu*
*Tulane University, USA*

**Xue Lin**                                                             *xue.lin@northeastern.edu*
*Northeastern University, USA*

**Min Xu**                                                              *mxu1@cs.cmu.edu*
*Carnegie Mellon University, USA*
*Mohamed bin Zayed University of Artificial Intelligence, UAE*

**Qifan Wang**                                                          *wqfcr@meta.com*
*Meta AI, USA*

**Tianyang Wang**[†]                                                    *tw2@uab.edu*
*University of Alabama at Birmingham, USA*

**Cheng Han**[†]                                                        *chk9k@umsystem.edu*
*University of Missouri-Kansas City, USA*

**Reviewed on OpenReview:** *https://openreview.net/forum?id=UwtXDttgsE*

---

[*]Equal contribution.
[†]Corresponding author.

## Abstract

In computer vision, Visual Prompting (VP) and Visual Prompt Tuning (VPT) have recently emerged as lightweight and effective alternatives to full fine-tuning for adapting large-scale vision models within the "pretrain-then-finetune" paradigm. However, despite rapid progress, their conceptual boundaries remain blurred, as VP and VPT are frequently used interchangeably in current research, reflecting a lack of systematic distinction between these techniques and their respective applications. In this survey, we revisit the designs of VP and VPT from first principles, and conceptualize them within a unified framework termed Prompt-based Adaptation (PA). Within this framework, we distinguish methods based on their injection granularity: VP operates at the pixel level, while VPT injects prompts at the token level. We further categorize these methods by their generation mechanism into fixed, learnable, and generated prompts. Beyond the core methodologies, we examine PA's integrations across diverse domains, including medical imaging, 3D point clouds, and vision-language tasks, as well as its role in test-time adaptation and trustworthy AI. We also summarize current benchmarks and identify key challenges and future directions. To the best of our knowledge, we are the first comprehensive survey dedicated to PA's methodologies and applications in light of their distinct characteristics. Our survey aims to provide a clear roadmap for researchers and practitioners in all area to understand and explore the evolving landscape of PA-related research.

## 1 Introduction

Large-scale vision models, exemplified by the Vision Transformer (ViT) (Dosovitskiy et al., 2021) and Swin Transformer (Liu et al., 2021), have fundamentally transformed computer vision. These models are typically pretrained on massive datasets (*e.g.*, ImageNet-21k (Russakovsky et al., 2015)) to acquire transferable representations, which can subsequently be finetuned for specific downstream tasks (Iofinova et al., 2022) (*e.g.*, FGVC (Jia et al., 2022), VTAB-1k (Zhai et al., 2019)). This approach is commonly referred to as the "pretrain-then-finetune" paradigm, which can markedly reduce the reliance on labeled data (Han et al., 2024). As the scale of these models continues to grow (Han et al., 2023), conventional full fine-tuning (FT), which updates all parameters, has become increasingly costly in terms of computation and storage, and risks eroding valuable pretrained knowledge Han et al. (2024). In response, a variety of parameter-efficient fine-tuning (PEFT) methods, aiming to finetune models by adjusting only a small fraction of parameters while keeping the remainder frozen, have been developed. Among these, Prompt-based Adaptation (PA) has emerged as a particularly prominent and effective technique (Jia et al., 2022).

In this survey, we provide a systematic review and categorization of recent PA algorithms and their practical implementations. Unlike existing surveys, which primarily focus on multimodal or vision–language settings, our work centers exclusively on PA within vision models. Understanding the confusing definitions of PA in the current research community, the primary contribution of this survey is to establish the ***first structured and unified overview of PA in large vision models***. We introduce a comprehensive taxonomy that first conquers numerous prompt-related research in large vision models into a unified scope, and then, in detail, divides them based on their distinct algorithmic designs and usages.

Our work is structured as follows. In §2, we begin by defining the overarching discipline of PA as the process of designing inputs at different locations to finetune a model's behavior. Within this field, we distinguish between two core paradigms in the visual domain: ❶ **Visual Prompting (VP)** and ❷ **Visual Prompt Tuning (VPT)**. In §2.2–2.3, we present the algorithmic foundations of VP and VPT, respectively, highlighting their related yet distinct perspectives toward achieving parameter efficiency. This categorization is determined by the geometric placement of prompts, distinguishing between those that modify the model's input and those that are integrated internally prior to the layer(s). In §2.4, we discuss the scopes of efficiency that PT and VPT focus on. In §3, we include PA's applications on foundational computer visions tasks, such as segmentation, restoration and enhancement, and compression. In §4, we explore the expanding

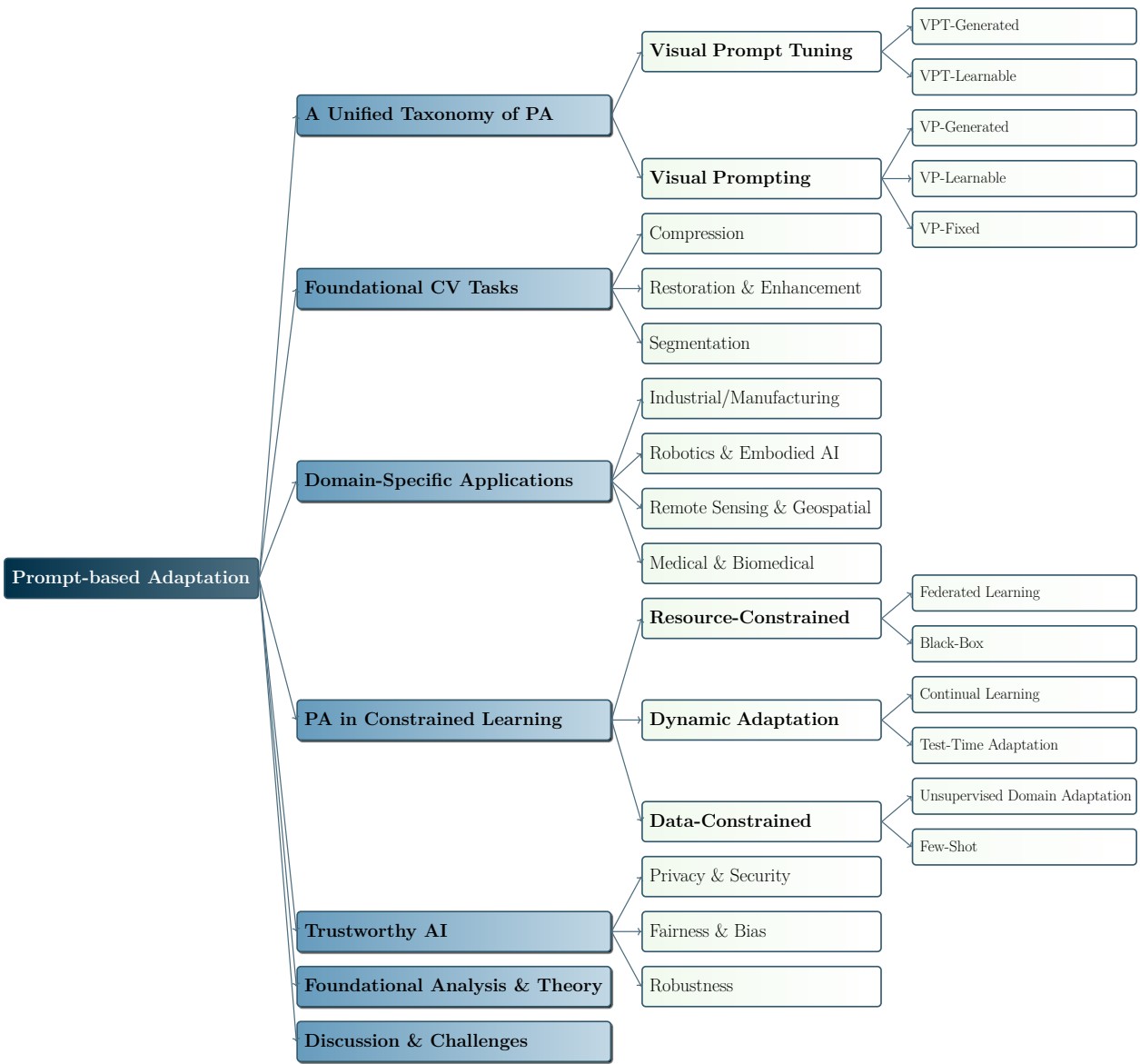

Figure 1: **A taxonomy of Prompt-based Adaptation (PA) in Large Vision Models.**

applications of PA across advanced machine learning problems and in various domain-specific contexts, such as medical imaging and robotics. In §5, our survey indicates that PA successfully demonstrates effectiveness in various scenarios with optional constraints. In §6, we discuss PA *w.r.t.* trustworthy, specifically categorizes into robustness, fairness and bias mitigation, and privacy and security. In §7, we delve into the foundational analysis and theoretical underpinnings of PA. Last but not least, in §8, we discuss key challenges and identify PA's promising future directions. The discussion encompasses pressing issues that remain to be addressed in the PA community, including safety considerations, training and inference latency, stability, and obstacles to real-world deployment. Acknowledging that PA has already been utilized in real-world scenes, the discussions are particularly valuable for guiding future research.

**Related Works.** Existing surveys related to PA focus on limited scopes, as they mainly focus on multimodal or vision–language settings. For example, Wu et al. (2024c) focuses on visual prompting in MLLMs, organizing techniques around visual instructions, prompt generation, and compositional reasoning, without covering internal pixel/token injections or parameter-efficient tuning in vision encoders. (Gu et al., 2023) provides a

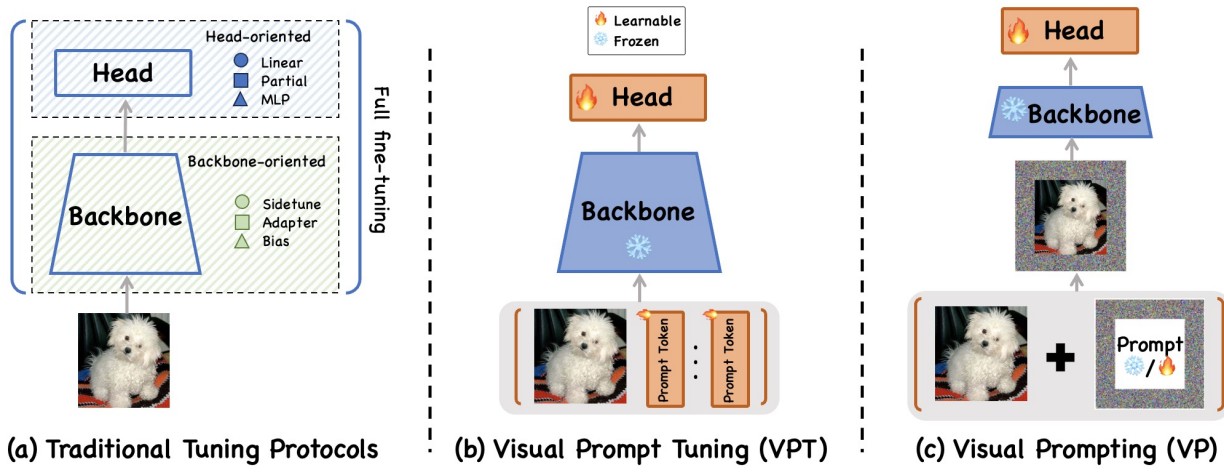

Figure 2: **Comparison of transfer learning and prompt-based adaptation methods.** (a) Current transfer learning protocols are grouped by tuning scope: Full fine-tuning, head-oriented, and backbone-oriented approaches. (b) Visual Prompt Tuning (VPT) freezes the backbone and optimizes additional prompt tokens together with the head. (c) Visual Prompting (VP) instead modifies the input space by adding prompts (which can be fixed or learnable), while keeping the backbone frozen and training a lightweight head.

systematic review of prompt engineering on vision–language foundation models (*e.g.*, CLIP/Flamingo/Stable Diffusion), emphasizing text-side prompts and VL pipelines rather than PA mechanisms inside vision backbones. (Lei et al., 2024) surveys prompt learning in computer vision from an AIGC-centric perspective grounded in VLMs and generative models, but do not unify methods by injection granularity or constrained-paradigm deployment. *Model Reprogramming: Resource-Efficient Cross-Domain Machine Learning* (Chen, 2022) provides an early theoretical foundation for VP by framing pixel-space transformation as a learnable input reprogramming layer for cross-domain transfer, inspiring later parameter-efficient prompting methods. More recently, (Ye et al., 2025) extends the discussion to large vision and multimodal models, tracing the evolution of VP techniques from pixel-level manipulation to foundation-level adaptation. In contrast, noticing the distinct differences on text-side and vision-side prompt-related attempts, we center on PA specifically on vision models, and propose a unified taxonomy that defines and disentangles previously ambiguous PA definitions. We further classify methods by generation mechanism (*i.e.*, learnable/generative/non-learnable) and injection granularity (*i.e.*, pixel- *vs.* token-level). Beyond methodology, we systematize constrained learning paradigms (*i.e.*, few/zero-shot, TTA, continual, black-box, forward-only, federated), consolidate domain applications (*i.e.*, medical, remote sensing, robotics, industrial), and add foundational analyses (*i.e.*, behavioral evidence and efficiency/theory), offering a deployment-oriented guidance not covered by prior surveys.

## 2 A Unified Taxonomy of Prompt-based Adaptation in Large-scale Vision Models

This section presents a taxonomy (see Figure 1) for prompt-based adaptation in large vision models. To avoid confusion, we separate where a prompt acts from how it is obtained. To provide a clear overview of the methods discussed, we summarize representative works for both VP and VPT in Table 1. Specifically, a method is considered representative if it is: (1) a pioneering work that established a key paradigm; (2) a canonical example of an algorithmic sub-type; or (3) a notable variant that demonstrates the field's diversity.

### 2.1 Preliminary on Large-scale Vision Models

Let $\mathbf{x} \in \mathbb{R}^{H \times W \times C}$ be an image. A frozen pre-trained vision encoder $f_\phi$ maps $\mathbf{x}$ to a representation that we view as a sequence (or grid) of features $Z^{(0)} \in \mathbb{R}^{T \times d}$. Here $T$ denotes the number of "sites": patch tokens for Transformers (Dosovitskiy et al., 2021; Liu et al., 2021), spatial cells for ConvNets (He et al.,

2016; Liu et al., 2022), or state steps for state-space encoders (Gu & Dao, 2023; Zhu et al., 2024b). The encoder consists of $L$ stacked blocks and produces $Z^{(L)}$; a task head $h_\omega$ (*e.g.*, a linear classifier, detector, segmentation head) outputs the prediction $\hat{\mathbf{y}} = h_\omega(Z^{(L)})$. In prompt-based adaptation we freeze $\phi$ and train only prompt parameters and, optionally, $\omega$.

## 2.2 Visual Prompting (VP): Input-space Prompting

VP modifies the input before tokenization/feature extraction via a prompt function $u(\,\cdot\,;\theta)$ (*e.g.*, (Bahng et al., 2022)):

$$\tilde{\mathbf{x}} = u(\mathbf{x};\theta), \qquad \hat{\mathbf{y}} = h_\omega\big(f_\phi(\tilde{\mathbf{x}})\big). \tag{1}$$

In general, VP can be further categorized into three different approaches: VP-Fixed, VP-Learned, and VP-Generated, based on how these prompts are generated:

- **VP-Fixed** introduces no learnable $\theta$ (*e.g.*, points/boxes/masks in interactive segmentation), thus its formulation remains identical to Eq. 1. These prompts are provided by rules or users without training. Typical forms are points, boxes, or masks for interactive segmentation (*e.g.*, SAM) (Kirillov et al., 2023) and simple visual/text hints for VLMs. These prompts are intuitive and zero-shot friendly, but their capacity is bounded by the prompt design space and the interface of the underlying model.
- **VP-Learnable** optimizes $\theta$ in pixel space (*e.g.*, overlays, masks, residuals, frequency cues) while keeping $\phi$ frozen (Bahng et al., 2022):

$$\min_{\theta,\,\omega} \mathbb{E}_{(\mathbf{x},\mathbf{y})}\Big[ \mathcal{L}\big(h_\omega(f_\phi(u(\mathbf{x};\theta))), \mathbf{y}\big) \Big] + \lambda\,\mathcal{R}(\theta). \tag{2}$$

  The learning process can be gradient-based (white-box), query-based (zeroth-order), or driven by small auxiliary modules. Fourier- or style-based prompts improve robustness and transfer under distribution shift. For medical segmentation, Fourier Visual Prompting (FVP) and Data-Dependent Frequency Prompt (DDFP) learn frequency-domain cues that regularize features across unseen domains (Wang et al., 2023c; Yin et al., 2025). OT-VP learns a universal visual prompt for target domains by aligning distributions with Optimal Transport (OT) (Zhang et al., 2025d), showing strong source-free or test-time adaptation performance (Zhang et al., 2025d). Local-Prompt introduces spatially local input prompts to reduce false positives in few-shot OoD detection (Zeng et al., 2024a). These methods keep adaptation external to the backbone yet deliver sizable gains under domain shift. Early "model reprogramming" shows that input-space patterns can repurpose black-box models with scarce data (Tsai et al., 2020). More recent works adopt zeroth-order or gradient-free updates to learn input prompts when gradients are unavailable. This line keeps the backbone untouched and fits API-only access patterns.
  For instance-level perception in satellite imagery, RSPrompter learns input prompts that guide instance segmentation with visual foundation models (Chen et al., 2024b). Promptable instance segmentation in remote sensing (*e.g.*, Insight Any Instance; ZoRI) leverages input cues to improve generalization and zero-shot recognition across scenes and sensors (Wang et al., 2024a; Huang et al., 2025a). In hyperspectral tracking, PHTrack and SPTrack inject spectral- or similarity-based input prompts that better exploit spectral structure for target localization (Chen et al., 2024c; Guo et al., 2024). In medical image segmentation, PASS performs test-time prompting to adapt styles and shapes without retraining the backbone (Zhang et al., 2024a).
  VP-Learned offers a simple, portable handle for robustness, OoD, and cross-domain transfer. It is effective when internal states are inaccessible or when we target low-cost deployment.
- **VP-Generated** utilizes a small generator $g_\psi$ to produce instance-adaptive prompts in input space (*cf.* black-box/instance-adaptive VP (Oh et al., 2023)):

$$\tilde{\mathbf{x}} = u\big(\mathbf{x}, g_\psi(\mathbf{x})\big) = (1-\mathbf{m})\odot\mathbf{x} + \mathbf{m}\odot\mathbf{r}_\psi(\mathbf{x}), \tag{3}$$

  where $\mathbf{m}$ is a spatial mask and $\mathbf{r}_\psi$ a synthesized residual/overlay. A canonical VP formulation casts prompting as inpainting: given masked input $(\mathbf{x},\mathbf{m})$, predict discrete visual tokens $\hat{z}_i$ and decode to pixels (Bar et al., 2022),

$$\hat{z}_i = \arg\max_{z_i} p_\theta\big(z_i \mid \mathbf{x}, \mathbf{m}\big), \quad \hat{\mathbf{y}} = \mathrm{Dec}(\hat{\mathbf{z}}), \tag{4}$$

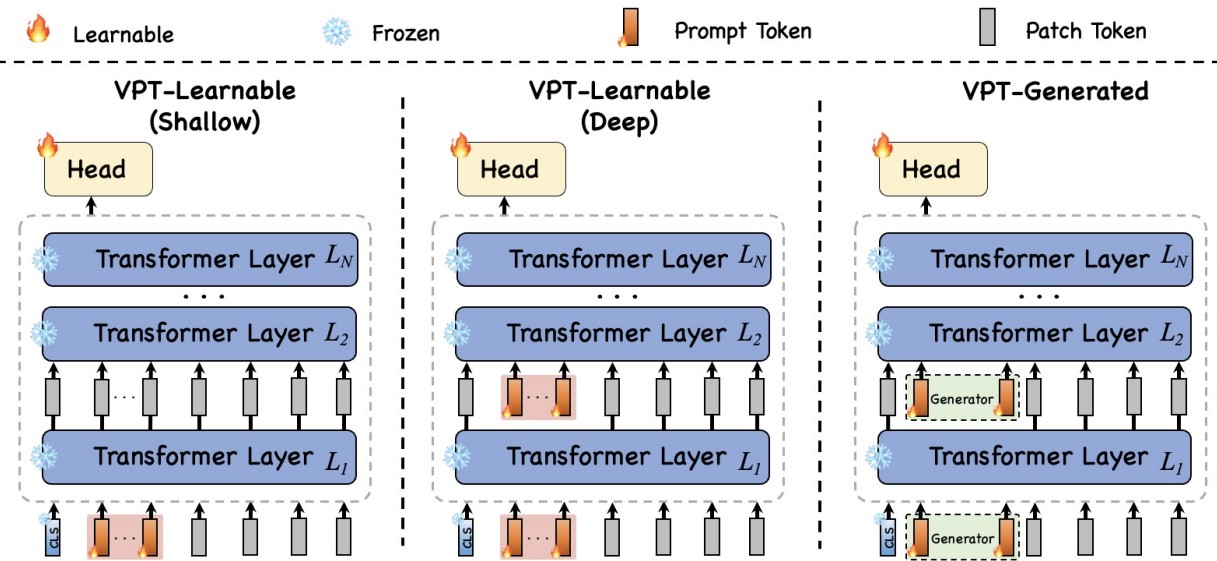

Figure 3: **Illustration of different variants of Visual Prompt Tuning (VPT, see §2.3)**. Left: **VPT-Learnable** (Shallow), where prompt tokens are only added at the first layer. Middle: **VPT-Learnable** (Deep), where prompt tokens are injected at every transformer layer. Right: **VPT-Generated**, where a generator produces instance-adaptive prompt tokens that are then inserted.

which operationalizes the prompt as a grid of input–output examples plus the query (all in pixel space). In black-box settings, $\theta$ (or $\psi$) can be updated with zeroth-order, as used in black-box visual prompting (Oh et al., 2023):

$$\widehat{\nabla}_\theta \mathcal{L} \;=\; \frac{\mathcal{L}(\theta + \alpha\Delta) - \mathcal{L}(\theta - \alpha\Delta)}{2\alpha}\,\Delta, \quad \Delta \sim \{\pm 1\}^{\dim(\theta)}. \tag{5}$$

Here, VP-Generated uses an auxiliary module to synthesize the input prompt per image (or per task), making the prompt instance-adaptive while remaining in pixel space. Typical generators include lightweight multi-layer perceptrons (MLPs) or small-scale CNNs.

For instance, BlackVIP builds a small-scale "coordinator" network that produces input-conditioned prompts; the coordinator is optimized with zeroth-order queries to a black-box model, enabling robust transfer without accessing internals (Oh et al., 2023). This design inherits the portability of VP and the flexibility of per-instance prompting, and it matches real API constraints. Other works synthesize per-image residuals or inpainting-style regions that steer the backbone at inference. They remain external to the model and are useful when gradients are not available or when we desire a unified input-facing adaptation across many backbones. Generated VP improves flexibility over static or global patterns while preserving the advantages of input-space control. It is a practical choice for black-box or multi-backbone ecosystems where a single input adaptor must generalize widely.

Note that for VP, all operations and categorizations focus on the pixel-level prompts. For internal injection prompts (see §2.3), even if learned or generated, they are defined as VPT since they act as learnable prompts into the token/feature sequence inside the network.

## 2.3 Visual Prompt Tuning (VPT): Internal Prompting

VPT injects learnable prompts into the token/feature sequence while freezing the backbone (Jia et al., 2022). While the majority of research explores VPT within the Transformer-based architectures, it is actually a general solution for various deep learning backbones (*e.g.*, ResNet (He et al., 2016), ConvNeXt-B (Liu et al., 2022)). In our survey, we adopt the attention layers in ViT as the primary example to illustrate VPT.

For a ViT with $L$ layers, patch embedding yields $X^{(0)} = [\mathbf{x}_{\text{cls}}^{(0)}; \mathbf{x}_1^{(0)}; \ldots; \mathbf{x}_N^{(0)}] \in \mathbb{R}^{(1+N)\times d}$. Following (Jia et al., 2022), two variants of VPTs can be distinguished based on the number of layers into which learnable prompts are incorporated. Specifically, VPT-Shallow prepends $p$ prompt tokens $P^{(0)} \in \mathbb{R}^{p\times d}$ only at the first layer (Jia et al., 2022):

$$Z^{(0)} = [\mathbf{x}_{\text{cls}}^{(0)}; P^{(0)}; \mathbf{x}_1^{(0)}; \ldots; \mathbf{x}_N^{(0)}], \quad Z^{(\ell+1)} = \text{Block}_\ell\big(Z^{(\ell)}\big), \; \ell = 0, \ldots, L-1. \tag{6}$$

VPT-Deep, on the other hand, uses layer-wise prompts $P^{(\ell)}$ by concatenating them at each layer's input (Jia et al., 2022):

$$Z_{\text{in}}^{(\ell)} = [\mathbf{x}_{\text{cls}}^{(\ell)}; P^{(\ell)}; \mathbf{x}_1^{(\ell)}; \ldots; \mathbf{x}_N^{(\ell)}], \quad Z^{(\ell+1)} = \text{Block}_\ell\big(Z_{\text{in}}^{(\ell)}\big). \tag{7}$$

Each Transformer block applies layer normalization multi-head self-attention (LN–MSA) and layer normalization multi-layer perceptron (LN–MLP) with residual connections:

$$\tilde{Z}^{(\ell)} = Z^{(\ell)} + \text{MSA}\big(\text{LN}(Z^{(\ell)})\big), \quad Z^{(\ell+1)} = \tilde{Z}^{(\ell)} + \text{MLP}\big(\text{LN}(\tilde{Z}^{(\ell)})\big). \tag{8}$$

Only the prompts and head are trained (Jia et al., 2022) during finetuning:

$$\min_{\{P^{(\ell)}\},\,\omega} \mathbb{E}_{(\mathbf{x},\mathbf{y})}\Big[ \mathcal{L}\big(h_\omega(\text{VPT}_{\{P^{(\ell)}\}}(f_\phi, \mathbf{x})), \mathbf{y}\big) \Big], \quad \phi \text{ frozen.} \tag{9}$$

From a methodology perspective, VPT can be categorized into VPT-Learnable and VPT-Generated:

- **VPT-Learnable** utilizes a small number of learnable prompt tokens added to the token sequence while keeping $\phi$ frozen (*cf.* Eqs. 6–9). These learnable prompts are optimized by gradient descent and can be inserted only at the first layer (*i.e.*, shallow) or at every layer (*i.e.*, deep) (Jia et al., 2022). Beyond the baseline VPT (*i.e.*, here stands for Jia et al. (2022)), many variants refine what tokens encode and how they are scheduled: For long-tailed classification, LPT adds class-aware prompt tokens and a re-weighted training schedule to balance head *vs.* tail classes (Dong et al., 2023). For self-supervised ViTs, improved token initialization/regularization stabilizes adaptation and reduces the gap to FT (Yoo et al., 2023). EXPRES builds learnable "output" tokens and residual prompt tokens to better steer frozen transformers, improving VTAB/FGVC benchmark performance with modest token counts (Das et al., 2023). SA$^2$VP learns a spatially aligned 2D map of prompt tokens and adapts them across depths via cross-attention, yielding stronger transfer on dense/classification tasks (Pei et al., 2024). VFPT augments prompt tokens with Fourier components to capture frequency cues, improving robustness under distribution shift with low parameter (Zeng et al., 2024b). SPT provides design heuristics on token length, placement, and initialization that consistently lift standard VPT (Wang et al., 2024b). E$^2$VPT introduces key–value prompt tokens and pruning to cut parameters/FLOPs while retaining accuracy, scaling from shallow to deep injection (Han et al., 2023). Adaptive Prompt tunes the prompt schedule (length/placement) with simple rules or meta-updates to reduce manual search (Le et al., 2025). DA-VPT (semantic-guided) aligns the distribution of prompt tokens with class semantics via metric learning, improving generalization across tasks (Ren et al., 2025).
- **VPT-Generated** uses a lightweight generator (*e.g.*, an MLP or a hypernetwork) to produce **prompt tokens** conditioned on the input or task, which are then inserted shallowly or layer-wise (*cf.* Eqs. 6–7). The motivation of this design is to improve prompts' instance adaptivity and reduce the manual and rigid design of their layouts. Instance-adaptive designs generate a token set per image: DVPT for medical analysis uses a bottleneck and cross-attention to derive sample-specific queries before emitting prompt tokens (He et al., 2025a); another DVPT variant employs a Meta-Net to produce a unique token for each image across recognition tasks (Ruan & Wang, 2023). ViaPT generates instance-aware prompt tokens for each image, enabling the model to better capture intra-class diversity while keeping the backbone frozen (Xiao et al., 2025c). Long-horizon conditioning can also drive token generation: LSPT gates information from earlier blocks to synthesize long-term spatial prompt tokens for self-supervised ViTs (Mo et al., 2024). Beyond recognition, a prompt token generator improves generative transfer in ViT-based synthesis (Sohn et al., 2023). Prompt Generation Networks (PGN) learn per-sample prompts with a tiny network; the generator itself operates in the latent/token space and integrates with frozen ViTs (Loedeman et al., 2024). Overall, the extra computational cost is the parameters and FLOPs of $g_\psi$; accuracy–efficiency is governed by the depth of injection (*i.e.*, shallow *vs.* deep) and the number of generated tokens.

## 2.4 Efficiency in Practice

The overall memory usage of fine-tuning a model can be divided into four parts: (1) *Model memory*, the storage of parameters; (2) *Activation memory*, the cache of intermediate features during the forward pass; (3) *Gradient memory*, the storage of gradients during backpropagation; and (4) *Optimizer memory*, the additional states maintained by optimizers (*e.g.*, momentum and variance in Adam). In full fine-tuning, all four components scale with the backbone size.

**Efficiency of VPT.** VPT freezes the backbone and updates only a small set of prompt tokens and the task head, so parameter gradients and optimizer states are allocated only for these lightweight modules. However, backpropagation must still traverse the entire backbone to compute token gradients, meaning that activation memory, which typically dominates the total GPU usage, remains largely unchanged. Thus, VPT effectively reduces the parameter and optimizer footprint but marginally alleviates activation-related memory cost (*e.g.*, full fine-tuning a vision backbone with hundreds of millions of parameters requires updating all weights, while VPT only introduces a small set of prompt tokens (*i.e.*, typically $< 0.5\%$ of parameters) that are optimized while the backbone remains frozen (Jia et al., 2022)).

While parameter-efficient approaches focus on reducing the number of trainable weights, a complementary line of work aims at minimizing activation memory during training. Recent memory-efficient fine-tuning (MEFT) (Kim et al., 2023; Simoulin et al., 2024) explores adaptive token or feature selection to avoid storing gradients for redundant activations, achieving substantial reductions in peak GPU memory with negligible performance degradation. These efforts suggest that parameter sparsity alone is probably insufficient for large-scale efficiency: activation optimization is becoming one of the key factors for scaling fine-tuning on commodity hardware.

To sum up, VPT still offers a highly efficient trade-off between accuracy and resource usage. By updating only a compact set of prompt tokens, it enables fine-tuning large-scale vision models on commodity GPUs and accelerates deployment in memory-constrained or latency-sensitive settings.

**Efficiency of VP.** VP operates at the input space and freezes the backbone by design. For *VP-Fixed* (*e.g.*, points/boxes/masks in SAM (Kirillov et al., 2023) or inpainting-style prompting (Bar et al., 2022)), there are *no trainable prompt parameters*, hence neither parameter gradients nor optimizer states are maintained for prompting itself; training-time memory reduces to the head (if any) and the activations required to backpropagate through the frozen backbone. For *VP-Learned* (Bahng et al., 2022) and *VP-Generated* (*e.g.*, BlackVIP (Oh et al., 2023), frequency- or distribution-aware variants such as FVP/DDFP (Wang et al., 2023c; Yin et al., 2025)), the trainable footprint remains lightweight ($\theta$ or a small generator $g_\psi$), so *parameter gradients and optimizer states are allocated only for these modules and the task head*, not for the backbone. However, as with VPT, backpropagation must still traverse the entire backbone to compute gradients *w.r.t.* $\theta$ or $g_\psi$, which implies that *activation memory largely remains*, typically the dominant component of peak GPU usage during training. Thus, VP effectively minimizes parameter/optimizer overhead (and eliminates it entirely in VP-Fixed), but only marginally alleviates activation-related cost.

In terms of practicality, VP offers two further efficiency advantages. First, VP-Fixed supports training-free or head-only adaptation, which *removes* prompt-side gradients and optimizer states by construction; the remaining memory is mostly due to activations through the frozen backbone and the small head, enabling extremely lightweight deployment settings. Second, VP is *black-box friendly*: gradient-free/zeroth-order optimization (as in BlackVIP (Oh et al., 2023)) avoids storing parameter gradients altogether, trading queries for memory and widening the feasibility on API-only backbones. During inference, composing prompts in the pixel space introduces negligible computational overhead (*e.g.*, border/overlay composition) relative to the main forward through the backbone; the runtime and memory are therefore dominated by the frozen encoder pass.

To sum up, VP provides a complementary efficiency profile to VPT: it *minimizes* parameter and optimizer states at the prompt side (to zero in VP-Fixed), preserves a frozen backbone, and enables black-box or training-free use cases, while leaving activation memory largely unchanged during training. This makes

Table 1: **Representative Visual Prompting (VP) and Visual Prompt Tuning (VPT) methods.**

|  | Method | Venue | Year | Sub-type | Prompt Space |
|---|---|---|---|---|---|
| **VP** | VPI (Bar et al., 2022) | NeurIPS | 2022 | Generated | Pixel |
|  | LabelMap (Chen et al., 2023) | CVPR | 2023 | Learnable | Pixel |
|  | SAM (Kirillov et al., 2023) | CVPR | 2023 | Fixed | Pixel |
|  | DAM-VP (Huang et al., 2023b) | CVPR | 2023 | Generated | Pixel |
|  | SSMask (Liu et al., 2024) | ICML | 2024 | Learnable | Pixel |
|  | InsVP (Zheng et al., 2024) | ACM MM | 2024 | Generated | Pixel |
|  | PixelVP (Sun et al., 2024c) | TMLR | 2024 | Learnable | Pixel |
|  | BayesVRP (Zhao et al., 2024b) | NeurIPS | 2024 | Learnable | Pixel |
|  | AttrVP (Chen & Wang, 2025) | ICLR | 2025 | Learnable | Pixel |
|  | LoR-VP (Jin et al., 2025) | ICLR | 2025 | Learnable | Pixel |
| **VPT** | VPT (Jia et al., 2022) | ECCV | 2022 | Learnable | Token |
|  | LPT (Dong et al., 2023) | ICLR | 2023 | Learnable | Token |
|  | EXPRES (Das et al., 2023) | CVPR | 2023 | Learnable | Token |
|  | SSL-VPT (Yoo et al., 2023) | ICML | 2023 | Learnable | Token |
|  | $E^2$VPT (Han et al., 2023) | ICCV | 2023 | Learnable | Token |
|  | VPT-Gen (Sohn et al., 2023) | CVPR | 2023 | Generated | Token |
|  | $SA^2$VP (Pei et al., 2024) | AAAI | 2024 | Learnable | Token |
|  | RePrompt (Wang et al., 2024b) | arXiv | 2024 | Learnable | Token |
|  | LSPT (Mo et al., 2024) | AAAI | 2024 | Generated | Token |
|  | VFPT (Zeng et al., 2024b) | NeurIPS | 2024 | Generated | Token |
|  | AdaPrompt (Le et al., 2025) | arXiv | 2025 | Learnable | Token |
|  | SG-VPT (Ren et al., 2025) | CVPR | 2025 | Learnable | Token |
|  | DVPT (He et al., 2025a) | NN | 2025 | Generated | Token |
|  | ViaPT (Xiao et al., 2025c) | ACM MM | 2024 | Generated | Token |
|  | SPT (Wang et al., 2024b) | ICML | 2024 | Learnable | Token |
|  | PAE (Wang et al., 2026) | ICLR | 2026 | Learnable | Token |

VP a practical choice for commodity hardware and latency-sensitive deployment, especially when API-only access or user-interactive prompting is required.

## 2.5 Comparative Analysis of Generation Mechanisms

Beyond efficiency trade-off, the selection among fixed, learnable, and generated prompting mechanisms is governed by the specific requirements of adaptation depth, data availability, and deployment constraints.

**Fixed Prompts.** This category, exemplified by the explicit points or boxes in SAM (Kirillov et al., 2023) and predefined templates in visual-language models, excels in *interactivity* and *interpretability*. Since these prompts are immutable during inference, they require no gradient updates, making them the only viable option for strictly black-box adaptation scenarios where model weights are inaccessible. However, their performance is strictly upper-bounded by the inherent zero-shot capability of the frozen backbone. They struggle to adapt to systematic domain shifts (*e.g.*, medical spectral bands) where the pre-trained feature space is misaligned with the target distribution, as they lack the capacity to recalibrate internal representations.

**Learnable Prompts.** Optimizing continuous vectors, as seen in VPT (Jia et al., 2022) and VP (Bahng et al., 2022), represents the standard paradigm for domain adaptation. By leveraging white-box gradient access, learnable prompts can bridge significant domain gaps (*e.g.*, ImageNet to Remote Sensing) by overfitting a lightweight set of parameters to the target distribution. Nevertheless, a critical limitation is their *static*

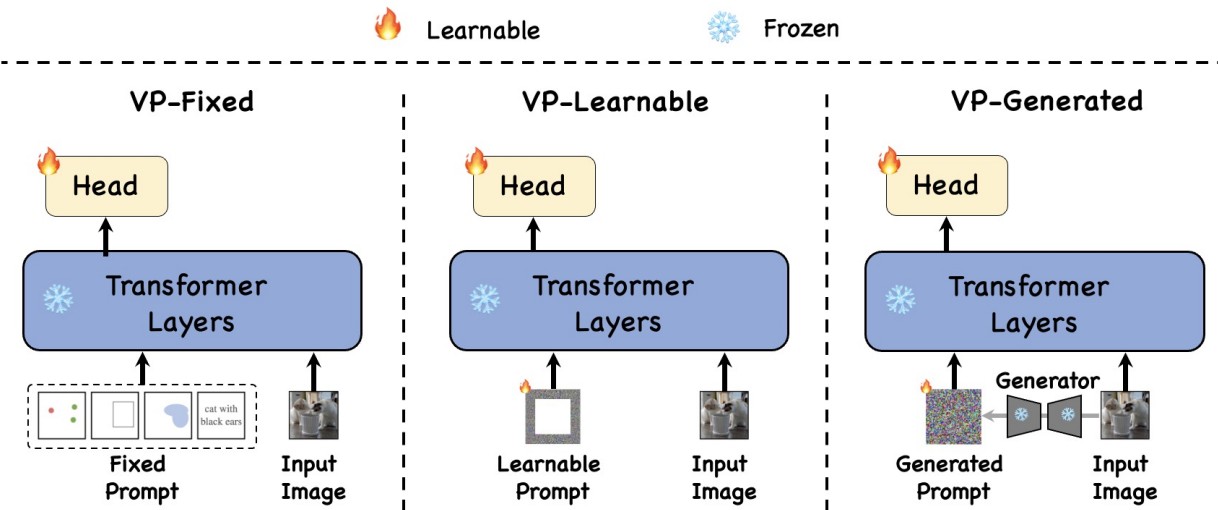

Figure 4: **Illustration of Visual Prompting (VP) variants (see §2.2).** (Left) **VP-Fixed**: prompts are predefined (e.g., boxes, points, text hints) and directly attached to the input without training. (Middle) **VP-Learned**: learnable prompts in the pixel space are optimized jointly with the task head while the backbone remains frozen. (Right) **VP-Generated**: a generator network produces instance-adaptive prompts for each input image, offering higher flexibility.

*nature*: once trained, the prompt remains identical for all input instances. This "dataset-level" optimization overlooks intra-class variations and instance-specific nuances, potentially limiting performance in highly diverse or open-world scenarios where dynamic adjustment is required.

**Generated Prompts.** To address the static limitation of learnable prompts, generated mechanisms (*e.g.*, IDPT (Zha et al., 2023), DAM-VP (Huang et al., 2023b)) introduce *instance-awareness*. By conditioning the prompt generation on individual input images via a lightweight network, these methods allow the model to dynamically adapt to varying lighting, scale, or occlusion conditions on a per-sample basis. This dynamic capability typically yields superior generalization and robustness compared to static learnable prompts. The trade-off, however, lies in the increased inference latency and optimization complexity, as the generator introduces an additional computational overhead and must be carefully regularized to prevent overfitting to the training distribution.

To sum up, fixed prompts offer the highest deployment flexibility; learnable prompts provide a strong baseline for stable domain transfer; and generated prompts offer the highest theoretical ceiling for complex, instance-varying tasks at the cost of computational complexity.

## 3 Foundational Computer Vision Tasks

We first introduce PA in foundational computer vision tasks (*i.e.*, segmentation (see §3.1), restoration and enhancement (see §3.2), compression (see §3.3), multi-modality (see §3.4)), aiming to provide a comprehensive guideline on how prompt mechanisms can bridge pre-trained large-scale vision models with downstream visual recognition and understanding.

### 3.1 Segmentation

PA methods are now being applied across a wide range of segmentation paradigms. In continual segmentation, *ECLIPSE* maintains a frozen backbone and adds visual prompts to update panoptic segmentation models, thereby mitigating catastrophic forgetting and ensuring stable plasticity over long task sequences (Kim et al., 2024a). For multimodal scenarios, *DPLNet* designs dual prompts for RGB and auxiliary modalities, fusing them with a lightweight module to minimize the number of trainable parameters (Dong et al., 2024a).

*GoPT* further group features into semantic clusters and inject prompts on a per-group basis, which improves cross-modal alignment using less than 1% of the total model weights for training (He, 2024).

Few-shot segmentation has benefited from class- and instance-aware prompting. *PAT* constructs prompts by transferring semantic cues from base to novel classes and by generating part-aware masks, an approach that enhances adaptation to new categories (Bi et al., 2024). *Hossain et al.* employ multi-scale prompting and causal attention between base and novel classes to strike a balance between generalization to new data and retention of prior knowledge (Hossain et al., 2024). Beyond the RGB domain, *Xu et al.* augment CLIP encoders with spectral prompts and utilize a spectral-guided decoder to improve pixel-level adaptation for unseen classes (Xu et al., 2024a). Prompting has also proven useful for low-level structural segmentation. *EVP* formulates explicit prompts by combining features from frozen patch embeddings with high-frequency components; this method unifies camouflaged object, forgery, shadow, and defocus-blur segmentation within a single framework. To address domain shifts in annotation style or image statistics, domain-adaptive prompting for SAM (*DAPSAM*) demonstrates that even interactive models can be steered by prompts learned in a new domain without altering the main model weights (Zhang et al., 2024d). Finally, VPT-style token prompts can be spatially aligned or implemented in a layer-wise fashion to better suit dense prediction tasks; $SA^2VP$, for example, aligns a 2D map of prompt tokens across network depths, improving both semantic and panoptic segmentation with few additional parameters (Pei et al., 2024).

## 3.2 Restoration and Enhancement

Image restoration must contend with a wide variety of degradations. PA offers a lightweight and flexible mechanism for encoding degradation-specific cues without re-engineering the entire network architecture. For all-in-one restoration, *PromptIR* injects degradation-aware prompts that guide a single backbone model across denoising, dehazing, and deraining tasks (Potlapalli et al., 2023). *PromptRestorer* extends this concept by extracting raw degradation features and processing them through a dedicated prompting branch with global–local modulation and gated propagation. This approach stabilizes training and elevates restoration quality across four distinct tasks (Wang et al., 2023a).

Frequency-aware prompts are effective for recovering fine details that spatial-only cues might miss. *FPro* decomposes features into frequency bands and utilizes dual prompt blocks to modulate low and high frequencies independently, demonstrating improved performance on five restoration tasks (Zhou et al., 2024b). Diffusion-based approaches can also be guided by prompts: *Diff-Restorer* leverages priors from Stable Diffusion and employs visual prompts to tackle a range of degradations within a unified framework (Zhang et al., 2024e). For compressed images, *PromptCIR* learns distortion-aware prompts that adapt to varying compression levels and artifacts, enhancing both fidelity and perceptual metrics (Li et al., 2024a). Beyond these methods, "ingredient-oriented" prompting encodes specific task factors as "ingredients," allowing a single model to dynamically switch behaviors as needed (Gao et al., 2023). In summary, these results point to a consistent pattern: by learning a small set of prompts that encapsulate what varies across conditions—be it the degradation type, frequency band, or compression level—it is possible to leave the core backbone frozen. This strategy keeps training computationally efficient while delivering substantial gains in restoration quality in practice.

## 3.3 Compression

PA is increasingly being applied to image and video compression, primarily to enhance adaptability (*e.g.*, enabling variable bit rates, region-of-interest (ROI) control, or user-defined quality preferences), without retraining large codec models end-to-end. One line of research conditions Transformer-based learned image compression with prompts, allowing a single model to cover multiple operating points. For variable-rate coding with ROI support, Kao *et al.* generate content-adaptive *prompt tokens* derived from the input image, an ROI mask, and a target rate. These tokens are then fed into the codec's Transformer blocks, effectively decoupling the decision of *what to preserve* (the ROI) from *how much to spend* (the bit rate) within a unified model (Kao et al., 2023). The same group further conditions the codec on user-selected *quality objectives* (*e.g.*, different perceptual metrics) via prompt tokens produced by a lightweight prompt generator, again avoiding the need for multiple specialized checkpoints (Kao et al., 2023).

Another approach adapts visual prompt tuning to modulate pretrained Transformer codecs. Qin *et al.* propose a layer-adaptive prompt module that injects prompts into both the encoder and decoder to steer attention and bit allocation across different rates. This method reduces parameter storage and data requirements while matching the rate–distortion performance of separately trained models (Qin et al., 2024). At a conceptual level, recent work on "rate–distortion–cognition (RDC)" argues that learned codecs should expose controllable parameters that balance human-perceived quality, machine utility, and bitrate; prompts serve as a natural mechanism for implementing such controls and conveying side information (Jia et al., 2024).

In the video domain, two complementary applications have emerged. First, prompts can drive *semantic* compression by aligning a compact representation with high-level tasks. Free-VSC learns prompts specific to vision foundation models to guide unsupervised video semantic compression without labels, optimizing the compressed stream for downstream understanding by vision–language models (Tian et al., 2024b). Similarly, SMC++ demonstrates that masked learning with prompt-like conditioning improves semantic coding pipelines and simplifies their engineering at scale (Tian et al., 2024a). Second, prompts can be applied on the *consumer* side: Compressed Video Prompt Tuning adapts models trained on raw video to compressed-domain inputs (*e.g.*, motion vectors, residuals) by re-parameterizing them into conditional prompts, which enhances recognition performance without retraining on raw pixels (Li et al., 2023a).

Prompted compression is also appearing at the interface of downstream tasks. UCIP utilizes dynamic prompts for universal compressed image super-resolution, improving restoration quality across different codecs and bitrates (Zhang et al., 2024b). For multimodal question answering, IQViC introduces a question-adaptive visual compressor that uses in-context text to determine *what content to preserve* before transmission, effectively transforming prompts into rate–content selectors (Yamao et al., 2024). Collectively, these results indicate a practical direction: prompts can function as a lightweight control plane for learned codecs (*i.e.*, dictating the rate, region, quality objective, or task utility), while the computationally intensive backbone remains frozen.

### 3.4 Multi-modal Tasks

Prompting is a cornerstone of interaction with multi-modal models, most notably with vision–language models (VLMs) such as CLIP (Radford et al., 2021). Early and foundational work largely explored *text prompts* (often called "prompt engineering") for recognition and retrieval. For example, *CoOp* learns continuous context vectors as class prompts on the text branch to improve few-shot classification (Zhou et al., 2022b); *CoCoOp* makes the learned text prompt conditional on the input image to generalize better across domains (Zhou et al., 2022a); and *MaPLe* jointly learns prompts on both the text and image branches for stronger cross-modal alignment (Khattak et al., 2023). Beyond prompt embeddings, other techniques have been explored: *CLIP-Adapter* adds a lightweight residual adapter to better fuse visual features with the text classifier (Gao et al., 2024), while *Tip-Adapter* uses cached features to build a classifier from class prototypes, avoiding full fine-tuning (Zhang et al., 2022). Similarly, open-vocabulary detection and segmentation use text queries as prompts (*e.g.*, *GLIP* (Li et al., 2022b)), and generative models are steered via textual prompts or learned text tokens (*e.g.*, textual inversion for diffusion models (Gal et al., 2023)). Expanding on the core PA philosophy of adapting the input guidance rather than the model architecture, recent works utilize Multimodal LLMs as scalable "AI-Experts" to steer generative models for specialized tasks. Wang et al. (2025a) exemplifies this by leveraging automated clinical feedback to optimize diffusion models via Direct Preference Optimization (DPO), effectively circumventing the bottleneck of human expert annotation.

More recently, the focus has shifted from text-side engineering to leveraging *visual prompts* to directly guide Multimodal Large Language Models (MLLMs). This paradigm allows for more intuitive and fine-grained control over model behavior. Users can provide explicit visual instructions through methods like free-form drawing, applying a set of marks or points to ground the model's attention (Yang et al., 2023a), or using one image as an exemplar to manipulate another (Yang et al., 2024b). Research in this area also explores how to best design these prompts for specific tasks, such as fine-grained control in image segmentation (Yang et al., 2023b) or enabling new paradigms for open-vocabulary detection (Zhang et al., 2023; Wu et al., 2025b). Beyond user-provided inputs, other works focus on automating the creation of visual prompts through cross-modal optimization, learning them in a training-free manner (Wu et al., 2024a), or jointly optimizing them

with text prompts. The transferability of these visual prompts (Jeong et al., 2025) and the integration of external knowledge (Lin et al., 2024b) are also active areas of investigation, highlighting the rapid growth of vision-side prompting in the multi-modal space.

**Scope of this Survey.** The methods above illustrate a rich and evolving landscape of multi-modal prompting. While we acknowledge the importance of both text-side and vision-side prompting for MLLMs, our survey's primary focus remains on the *internal mechanisms* of prompt-based adaptation (PA) on vision-centric backbones. VP and VPT, as defined in this survey, represent the core algorithmic primitives for adapting the vision encoder itself. We discuss multi-modal works to provide essential context and to clarify the distinction between adapting the VL interface versus adapting the vision pathway, which is our central theme.

## 4 Domain-Specific Applications

While PA has demonstrated remarkable success in conventional computer vision tasks, its full impact is best revealed through practical deployments across a range of heterogeneous, real-world domains. In this section, we provide a systematic overview of how PA is being adapted to diverse applied settings. We organize existing applications based on the unique demands and intrinsic challenges of the target environment. In the following subsections, we organize our discussion by application domain rather than strictly separating methods into VP and VPT categories. Since the adoption of these paradigms is not uniformly balanced across all fields; some domains currently favor one approach over the other. Therefore, while we comprehensively discuss how PA addresses domain-specific challenges, we refer readers to Table 2 and Table 3 for comprehensive method-level classifications of VP and VPT applications, respectively.

Current domain-specific uses of PA can be categorized into four major areas: medical and biomedical imaging (see §4.1), remote sensing and geospatial analysis (see §4.2), robotics and embodied AI (see §4.3), industrial inspection and manufacturing (see §4.4), autonomous driving and advanced driver-assistance system (ADAS) (see §4.5), 3D Point Clouds and LiDAR (see §4.6), video understanding and temporal perception (see §4.7), and underwater and adverse Environments (see §4.8). These categories are defined not by PA architectural variations but rather by the distinct operational contexts and domain-specific challenges they present.

### 4.1 Medical and Biomedical Imaging

Medical and biomedical imaging represents a critical domain where PA has demonstrated compelling utility, driven by the fundamental need for data efficiency, cross-modality generalization, and interpretability. PA offers a modular approach for adapting large-scale vision models to segmentation, classification, and reporting tasks across various imaging modalities (*e.g.*, CT, MRI, X-ray, histopathology), which frequently operate under limited supervision (*i.e.*, data scarcity) and strict clinical constraints (Xiao et al., 2024; 2025d; Wei et al., 2025). A prominent direction is the adaptation of vision foundation models, such as the *Segment Anything Model (SAM)* (Kirillov et al., 2023), to medical segmentation tasks via learnable visual prompts. Works like *Customized SAM* and *3D SAM-Adapter* extend SAM to radiological and volumetric datasets by injecting 2D or 3D spatial prompts that encode lesion or organ priors (Zhang & Liu, 2023; Gong et al., 2024). *Ma-SAM* further refines this paradigm by introducing a modality-agnostic prompt encoder capable of handling multimodal volumetric data through the joint optimization of spatial and semantic cues (Chen et al., 2024a). These designs exploit the structure-preserving benefits of pixel-based prompting while leveraging the semantic scalability of foundation models like SAM.

Beyond segmentation, VPT has also been applied to clinical report generation and multimodal reasoning. *PromptMRG* leverages diagnosis-driven prompts to align imaging features with clinical report templates, thereby improving factual alignment and coherence in generation (Jin et al., 2024). Concurrently, *Biomed-DPT* employs dual-modality prompt tuning to bridge vision-language pretraining with medical text, enabling few-shot adaptation for diverse downstream tasks such as classification, grounding, and captioning (Peng et al., 2025). This line of work highlights how prompt design can function as a conduit for clinical knowledge integration. To address distribution shifts across devices, sites, or patient populations, VPT can be applied for domain generalization and source-free adaptation. For instance, *FVP* and *DDFP* utilize frequency-domain prompts to regularize representations across domains, demonstrating strong robustness under unseen

test distributions (Wang et al., 2023c; Yin et al., 2025). *ProSFDA* integrates prompt learning into source-free domain adaptation pipelines, achieving improved alignment without requiring access to source data at test time (Hu et al., 2022).

Finally, real-world deployments in federated and privacy-sensitive settings have inspired task-specific prompting strategies. *FedLPPA* proposes personalized prompt learning across decentralized clients for weakly-supervised segmentation, while *PASS* applies test-time visual prompting to adapt styles and shape priors under distribution drift (Lin et al., 2024a; Zhang et al., 2024a). SafeTriage leverages prompt-based anonymization to preserve identity protection in facial video triage while retaining task relevance (Savic & Zhao, 2023). Meanwhile, recent benchmarking efforts such as *A Real-World Dataset* provide standardized datasets for evaluating foundation model adaptation across real-world hospitals and imaging systems (Wang et al., 2023b). Together, spanning learnable tokens, pixel-space injection, and cross-modal generation, these advances reveal how VPT methods are actively reshaping medical imaging workflows. By offering a parameter-efficient, interpretable, and context-aware adaptation mechanism, VPT is poised to bridge the gap between foundation models and the safety-critical demands of clinical AI systems.

In the medical domain, the choice between VP and VPT is largely dictated by the task modality and interaction requirements. VP, particularly in its fixed and generated forms, dominates segmentation tasks. This preference stems from the inherent compatibility between pixel-space prompts (*e.g.*, points, boxes) and clinical workflows, where human-in-the-loop interaction is required to localize lesions using foundation models like SAM. Conversely, VPT prevails in high-level reasoning tasks such as report generation and disease classification. By injecting learnable tokens directly into the feature space, VPT facilitates the integration of specialized clinical knowledge and multi-modal alignment (*e.g.*, image-to-text) while maintaining the structural integrity of the frozen backbone, a critical advantage for data-efficient adaptation in resource-constrained or federated medical environments.

## 4.2 Remote Sensing and Geospatial Analysis

Remote sensing and geospatial analysis present unique challenges, such as domain heterogeneity, extreme class imbalance, and spectral complexity (Cheng et al., 2020). Consequently, PA has been increasingly explored as a mechanism to efficiently adapt foundation models for tasks such as segmentation, retrieval, and change detection, particularly under data-scarce or multi-modal conditions.

A central theme in this area is the adaptation of foundation models for object detection and instance segmentation. RSPrompter, for instance, utilizes task-specific visual prompts to guide instance segmentation in satellite imagery, improving precision and instance separation without full model retraining (Chen et al., 2024b). Insight Any Instance and ZoRI extend this concept to generalizable and zero-shot segmentation tasks, where instance-level visual cues are injected via learnable prompts to enhance detection robustness across diverse scenes and sensors (Wang et al., 2024a; Huang et al., 2025a; Xiao et al., 2025b; Wang et al., 2025c). Prompting-based approaches have also proven beneficial for temporal reasoning tasks such as change detection and captioning. *A Decoupling Paradigm* employs prompt tuning to disentangle static and dynamic scene features, enabling the accurate description of semantic changes over time (Liu et al., 2023). Similarly, VPT-enhanced CLIP has been used to guide bi-temporal change detectors with learned priors that highlight expected spatial transformations, thereby improving generalization under noisy labels (Liu et al., 2025c; Ji et al., 2025). Cross-modal retrieval represents another important use case where prompts serve to align vision and language spaces for effective geospatial understanding. Methods like *Parameter-efficient transfer* and *RLita* explore lightweight, prompt-based fine-tuning for satellite-to-caption and caption-to-region alignment, outperforming full model tuning while maintaining modality consistency (Yuan et al., 2023; Zhang et al., 2025b). In the context of few-shot classification, works such as *MVP* and the *Few-shot Survey* demonstrate that VPT can encode spatial priors or domain-specific knowledge, thereby improving performance on unseen classes and datasets with minimal supervision (Zhu et al., 2024a; Qiu et al., 2024). For broader foundation model adaptation, methods like *UPETU* and *LayerLink* introduce parameter-efficient prompt encoders capable of tuning pretrained vision backbones for scene classification and detection without requiring access to the full training data or GPU-intensive backpropagation (Dong et al., 2024b; Zhu et al., 2025). These efforts underscore the scalability of prompt-based fine-tuning in real-world Earth observation.

Finally, hyperspectral tracking and classification have recently gained attention through spectral-aware prompting. *PHTrack* and *SPTrack* integrate domain-specific spectral similarity prompts or spatial priors to guide transformer backbones, enabling them to better utilize rich spectral information in hyperspectral video and image tracking tasks (Chen et al., 2024c; Guo et al., 2024). In summary, these diverse applications demonstrate the utility of VPT for supporting flexible and modular model adaptation in remote sensing. It bridges data modality gaps, enhances domain robustness, and reduces computational overhead, positioning it as a promising paradigm for scalable Earth intelligence systems.

Analyzing the adaptation strategies in remote sensing reveals a clear division of labor driven by the unique characteristics of geospatial data. VP has become the go-to strategy for precision-oriented tasks, such as instance segmentation and object tracking (Chen et al., 2024b; Guo et al., 2024). The rationale is straightforward: satellite imagery is notoriously dense and cluttered; explicit pixel-level prompts (whether spatial or spectral) act as strong guidance signals, helping frozen models isolate specific targets amidst complex backgrounds where generic features might fail. In contrast, VPT finds its niche in high-level semantic tasks like scene classification and cross-modal retrieval (Zhang et al., 2025b; Zhu et al., 2024a). Given the massive visual gap between standard ground-level pre-training data (*e.g.*, ImageNet) and the "bird's-eye" view of remote sensing, injecting learnable tokens deep into the network architecture proves to be a more effective way to realign the global feature space for this distinct domain, all while bypassing the prohibitive cost of full-parameter retraining.

### 4.3 Robotics and Embodied AI

Robotics and embodied AI represent a rapidly evolving frontier where PA plays a critical role in bridging large-scale foundation models with the complex demands of 3D perception, motion reasoning, and embodied interaction. In the context of 3D understanding, prompts serve as a lightweight mechanism for adapting 2D pretrained models to point cloud data. Methods such as PointCLIP V2 Zhu et al. (2023) and CLIP2Point Huang et al. (2023c) align CLIP embeddings with point cloud features through contrastive prompting, while P2P Wang et al. (2022d) introduces point-to-pixel prompting that enables the direct transfer of pretrained vision-language knowledge to 3D point clouds. Further research from GAPrompt Ai et al. (2025) and PointLoRA Wang et al. (2025b) demonstrates that geometry-aware or low-rank prompts can outperform full fine-tuning in open-world and few-shot tasks. Techniques such as instance-aware dynamic prompts Zha et al. (2023) and other parameter-efficient methods like Point-PEFT Tang et al. (2024a) expand the utility of prompts for task specialization across various 3D recognition benchmarks. For embodied interaction, architectures like ShapeLLM Qi et al. (2024) and Any2Point Tang et al. (2024b) extend PA frameworks to support multi-modal grounding Song et al. (2025) and affordance-based reasoning for manipulation tasks. Canonical shape prompting facilitates few-shot, class-incremental 3D learning by generating view-invariant prototypes Cheraghian et al. (2024). In the unified driving perception, visual exemplar-driven task prompting enables label-efficient scene parsing across heterogeneous sensor modalities Liang et al. (2023).

Within the video domain, prompt tuning has been shown to support temporal generalization and class-incremental learning. Space-Time Prompting introduces dynamic temporal prompts for continual video recognition Pei et al. (2023), while SimDA integrates diffusion models with prompt-driven adapters for efficient video generation Xing et al. (2024). Additionally, Zeroi2V proposes a zero-cost adaptation scheme to repurpose pretrained image transformers for video understanding, highlighting the synergy between prompt-based temporal transfer and frozen backbones in embodied AI scenarios Li et al. (2024b). Collectively, these studies illustrate that VPT serves as a flexible and parameter-efficient interface for robotics and embodied AI applications. Prompts can be used to condition models on structural priors (*e.g.*, shape, motion), guide modality fusion, and support continual adaptation in dynamic, real-world environments. As robotics increasingly relies on multi-modal and 3D inputs, prompt-based learning offers a scalable pathway for bridging the gap between general-purpose foundation models and specialized embodied intelligence.

The deployment of PA in robotics reflects a strategic effort to bridge the dimensional gap between 2D pre-training and 3D physical reality. VP, particularly via input-level projection or decoration, serves as the primary bridge for adapting vision-language models to point clouds (Wang et al., 2022d; Zhu et al., 2023). By reformatting sparse 3D data into 2D-compatible views, VP allows robotic systems to exploit the vast, open-world semantic knowledge of models like CLIP without the prohibitive cost of curating massive

3D datasets. Conversely, VPT becomes indispensable when tasks demand deep physical reasoning, such as grasping geometry or tracking temporal motion (Ai et al., 2025; Pei et al., 2023). In these scenarios, learnable tokens act as carriers for structural priors, effectively injecting the missing spatio-temporal context into the backbone that static image pre-training inherently lacks.

## 4.4 Industrial Inspection and Manufacturing

PA is gaining traction in high-precision manufacturing for tasks such as defect detection, anomaly segmentation, and visual quality control. A common paradigm involves maintaining a frozen backbone and using prompts to steer powerful foundation models, such as SAM or CLIP, toward product-specific visual cues with minimal or no additional supervision. At the pixel level, several works adapt the Segment Anything Model (SAM) to industrial defects using promptable masks. An unsupervised pipeline for laser-based additive manufacturing generates pseudo-labels and region masks to localize porosity without manual annotation, demonstrating that promptable segmentation is viable even with noisy factory data (Era et al., 2023). SAA+ proposes *hybrid prompt regularization* to segment anomalies without any training, improving zero-shot transfer by stabilizing how prompts guide the model (Cao et al., 2023). Self-Perception Tuning (SPT) further incorporates a lightweight self-perception branch to refine SAM's masks at test time, enhancing industrial anomaly segmentation while keeping the core model frozen (Yang et al., 2025). Extending beyond per-image prompts, SAID introduces *scene prompts*, enabling a single model to segment diverse industrial defects under varying lighting and background conditions (Huang et al., 2025b). Another line of research combines CLIP and SAM, allowing semantic cues from CLIP to serve as spatial prompts for SAM. ClipSAM aligns textual and image-based cues in CLIP and uses them to constrain SAM's masks, yielding strong zero-shot anomaly segmentation performance on benchmarks like MVTec-AD and VisA (Li et al., 2025e). A two-stage CLIP–SAM framework similarly leverages CLIP for coarse anomaly localization before refining the results with SAM, thereby boosting segmentation quality on industrial datasets (Hou et al., 2024). Additionally, image-aware prompt generators can synthesize dynamic prompts on a per-sample basis to better adapt to fine-grained surfaces and textures prior to segmentation. Prompted anomaly detection with vision–language backbones is also an active area in industrial quality assurance. WinCLIP demonstrates that carefully designed prompt ensembles and windowed features can elevate zero- and few-shot anomaly classification and segmentation (Jeong et al., 2023). AnomalyCLIP learns *object-agnostic* text prompts that generalize across different product types for zero-shot detection (Zhou et al., 2024a). VCP-CLIP injects *visual context prompts* into the text encoder of CLIP, which reduces the need for product-specific prompt engineering and improves zero-shot anomaly segmentation across numerous real-world factory datasets (Qu et al., 2024). These methods are parameter-efficient and particularly practical for scenarios where only API access to the model is available.

While less common, token-level Visual Prompt Tuning (VPT) for industrial inspection is an emerging direction. For instance, masked prompt tuning atop self-supervised features has been explored to enhance surface defect inspection with minimal labels, showing that small sets of learnable prompts can match or exceed the performance of full fine-tuning under stringent memory constraints (Wu et al., 2025a). Overall, the trajectory is clear: prompt-based adapters (*i.e.*, whether in the pixel space (VP) or via lightweight prompt modules), offer a simple and deployable pathway for integrating foundation models into production quality assurance pipelines with low annotation cost and rapid iteration.

The industrial landscape presents a unique "rare event" challenge: defective samples are scarce, making traditional supervised training difficult. Consequently, VP has emerged as the dominant paradigm, particularly for zero-shot and few-shot anomaly detection (Jeong et al., 2023; Cao et al., 2023). By leveraging the open-vocabulary capabilities of CLIP or the zero-shot segmentation power of SAM, pixel-level or text-guided prompts allow systems to define "normality" and localize outliers without requiring exhaustive defect datasets. On the other hand, VPT is carving out a niche in surface inspection tasks involving complex textures (Wu et al., 2025a). Here, injecting learnable tokens helps fine-tune the backbone's sensitivity to subtle material variations (*e.g.*, scratches on metal versus fabric) that generic pre-trained features might overlook, offering a parameter-efficient alternative to full retraining in memory-constrained factory environments.

### 4.5 Autonomous Driving and Advanced Driver-Assistance System (ADAS)

The domain of autonomous driving demands exceptional robustness against severe domain shifts, such as variations in lighting (night *vs.* day), adverse weather conditions (rain and fog), sensor artifacts (lens blur), and changes in geographic location or hardware configurations. In this context, PA has emerged as a practical methodology for enhancing the performance of large vision models without necessitating complete backbone retraining. At the pixel level, visual prompting has been utilized to guide segmentation models in adverse weather conditions. Differentiable implicit prompts can be optimized end-to-end to improve road parsing on Cityscapes and related benchmarks under fog, rain, and low light (Kalwar et al., 2023). For scenarios where strong segmentation priors are available, prompting foundation models (*e.g.*, SAM-style pipelines) has proven beneficial; recent work shows that semantic prompts can transfer from Cityscapes to ACDC, Dark Zurich, and other adverse-condition datasets using simple input-level designs (Wang et al., 2024c). Furthermore, learning the prompts themselves, rather than relying on fixed prompt types, serves to close the performance gap with full fine-tuning while maintaining a frozen backbone (Huang et al., 2024). Token-level prompt tuning has been applied directly to driving benchmarks. One source-free domain adaptation study demonstrated that replacing backbone updates with visual prompt tuning enables successful adaptation from synthetic (GTA5/SYNTHIA) to real-world (Cityscapes) domains, revealing that per-layer prompt tokens can encapsulate the majority of the transfer signal even when the core network gradients are inaccessible (Ma et al., 2023). A follow-up work targeting adverse driving scenes proposes severity-aware prompt tuning that conditions prompts on weather intensity and employs an instructive chain-of-domain schedule; this approach improves semantic segmentation across multiple adverse domains without modifying the backbone weights. Bird's-eye-view (BEV) perception introduces unique adaptation challenges, as it inherently couples multi-camera geometry with semantic understanding. Xiao et al. (2026; 2025a) In few-shot BEV learning, visual prompts are employed to "warm-start" prediction heads from limited data, thereby reducing the dependency on extensively labeled frames for inferring road layouts and object cues. Cross-modal alignment methods also incorporate prompt mechanisms, such as shared prompts to link camera features with language supervision for BEV retrieval and segmentation (Xie et al., 2025b). The application of prompting has also extended to end-to-end systems. For camera-only driving, injecting learned tokens into multi-modal blocks helps stabilize the policy across varied scenes and weather conditions, incurring minimal parameter overhead and requiring no changes to the backbone encoders. Beyond perception and control, perspective-to-BEV instruction generation utilizes prompt tuning to condition a large model on urban context (*e.g.*, lanes, signs, topology) to produce navigation instructions; here, the core challenge is to formulate prompts that ensure consistency between the model's BEV priors and the camera-view inputs (Yang et al., 2024c).

Two distinct patterns emerge from these applications. First, simple pixel-space designs remain effective when model internals are inaccessible or gradients are unavailable, and they are readily deployable in black-box settings. Second, token-level prompt tuning tends to yield an optimal trade-off between adaptation and accuracy when gradients are available, particularly with per-layer prompts or severity-aware scheduling. Open issues include latency, as prompt generators can add overhead at scale; safety, as prompts must not trigger brittle behavior in rare, long-tail events; and calibration, since instance-adaptive prompts require uncertainty checks prior to deployment. The trajectory is evident: prompts are increasingly becoming the default mechanism for maintaining the robustness of autonomous systems as operational conditions evolve.

Ensuring operational resilience in non-stationary environments is the primary driver for PA in autonomous driving. Under these environments, weather, lighting, and sensor noise constantly fluctuate. VP serves as a compelling strategy for input-level rectification, particularly when dealing with adverse weather or when the perception module is a black box (Kalwar et al., 2023; Huang et al., 2024). In these cases, pixel-space prompts effectively act as a "preprocessing" layer that normalizes corrupted inputs before they reach the frozen backbone. Conversely, VPT is the preferred approach for addressing systemic domain shifts, such as Sim-to-Real transfer or cross-sensor BEV alignment (Ma et al., 2023). By accessing internal gradients, learnable tokens can recalibrate the deep feature space to bridge the gap between synthetic training data and real-world diversity, providing a more robust adaptation mechanism than surface-level pixel manipulation could achieve alone.

Table 2: **Comprehensive summary of VP methods across domains.**

| Domain | Method | Task | Sub-type |
|---|---|---|---|
| Medical | CusSAM (Zhang & Liu, 2023) | Spatial prompts for medical segmentation | VP-Learned |
| Medical | SAM-Ada (Gong et al., 2024) | 3D volumetric prompting for radiology | VP-Learned |
| Medical | Ma-SAM (Chen et al., 2024a) | Modality-agnostic prompt encoder for multimodal | VP-Learned |
| Medical | FVP (Wang et al., 2023c) | Frequency prompting for cross-domain generalization | VP-Learned |
| Medical | DDFP (Yin et al., 2025) | Data-dependent prompting for domain shift | VP-Learned |
| Medical | PASS (Zhang et al., 2024a) | Test-time prompting for style/shape adaptation | VP-Learned |
| RS | RSPrompter (Chen et al., 2024b) | Instance segmentation prompts in remote sensing | VP-Learned |
| RS | IA Instance (Wang et al., 2024a) | Generalizable zero-shot instance segmentation | VP-Learned |
| RS | ZoRI (Huang et al., 2025a) | Discriminative zero-shot RS recognition | VP-Learned |
| RS | PHTrack (Chen et al., 2024c) | Spectra prompting for hyperspectral tracking | VP-Learned |
| RS | SPTrack (Guo et al., 2024) | Spectral prompts for tracking and classification | VP-Learned |
| Industrial | Unsupervised AM (Era et al., 2023) | Promptable segmentation in additive manufacturing | VP-Fixed |
| Industrial | SAA+ (Cao et al., 2023) | Training-free prompt for anomaly segmentation | VP-Fixed |
| Industrial | SPT (Yang et al., 2025) | Self-perception tuning of SAM masks at test time | VP-Learned |
| Industrial | SAID (Huang et al., 2025b) | Scene prompts for industrial defect segmentation | VP-Learned |
| Industrial | ClipSAM (Li et al., 2025e) | CLIP-guided semantic/spatial prompts for SAM | VP-Generated |
| Industrial | CLIP→SAM (Hou et al., 2024) | CLIP coarse localization + SAM refinement | VP-Generated |
| Autonomous | DiffPrompter (Kalwar et al., 2023) | Differentiable implicit prompts for road segmentation | VP-Learned |
| Autonomous | SAMDA (Wang et al., 2024c) | Semantic prompts for adverse-weather datasets | VP-Fixed |
| Autonomous | SSPrompt (Huang et al., 2024) | Learnable pixel prompts replacing fixed prompts | VP-Learned |
| 3D | P2P (Wang et al., 2022d) | Point-to-pixel prompting for 3D transfer | VP-Learned |
| Underwater | WaterSAM (Hong et al., 2024) | LoRA + underwater segmentation prompts | VP-Learned |
| Underwater | USIS-SAM (Lian et al., 2024) | Saliency prompt generation for underwater imagery | VP-Generated |
| Underwater | UWSAM (Li et al., 2025c) | End-to-end underwater prompt generator | VP-Generated |
| Adverse | SAM-EDA (Wang et al., 2024c) | Semantic prompts for weather segmentation | VP-Learned |

### 4.6   3D Point Clouds and LiDAR

Data from 3D sensors like LiDAR, often represented as point clouds, exhibits a sparse, irregular, and permutation-invariant structure that fundamentally differs from the dense grid of 2D images. Despite these structural disparities, PA has proven transferable to this domain with only minor modifications to the model interface. Within this domain, two primary patterns have emerged: (i) input-space prompting, which reformats 3D signals for consumption by a frozen 2D or 3D encoder, and (ii) token-level prompting, which inserts learnable tokens directly into point cloud Transformers while the backbone remains fixed. One line of research focuses on adapting 2D pretrained models to 3D tasks by *prompting the input.* Point-to-pixel prompting, for instance, converts a point cloud into geometry-preserving renderings, enabling a frozen image model to be tuned for 3D tasks using only lightweight prompt parameters (Wang et al., 2022d). Related efforts transfer CLIP to point clouds via image–depth pretraining or multi-view rendering, subsequently attaching small adapters or prompts to facilitate few- or zero-shot recognition (Huang et al., 2023c; Zhu et al., 2023). These methods demonstrate that input-space prompting can effectively repurpose large 2D or vision–language encoders for point-cloud classification, segmentation, and detection without full fine-tuning. The second, and now mainstream, direction is *token-level prompting* for 3D Transformers. Instance-aware Dynamic Prompt Tuning (IDPT) generates sample-specific prompt tokens that account for point-level noise and shape variation, improving transfer over static prompts with a minimal parameter budget (Zha et al., 2023). Dynamic Adapter Meets Prompt Tuning (DAPT) further couples per-token adapters with internal prompt tokens while freezing the backbone, yielding strong accuracy–efficiency trade-offs on benchmarks like ScanObjectNN, ModelNet40, and ShapeNetPart. Beyond generic prompts, more sophisticated designs inject geometry-aware priors, such as surface normals or curvature, directly into the prompt stream; GAPrompt, for example, introduces a point-wise prompt branch and a propagation mechanism to encode global and

local geometry while maintaining a low count of trainable parameters (Ai et al., 2025). A complementary approach treats the positional encodings themselves as a form of prompt: positional prompt tuning revisits 3D positional codes to learn compact position prompts that efficiently aggregate multi-scale structure (Zhang et al., 2024c). To ensure stability across domains, Point-PRC regularizes the interaction between task-specific prompts and task-agnostic knowledge in large 3D backbones, thereby improving domain generalization without modifying frozen weights (Sun et al., 2024b). A practical strategy for maximizing parameter efficiency involves pairing prompts with low-rank adapters; PointLoRA demonstrates how LoRA modules, combined with token selection, can reduce trainable parameters while preserving performance on point cloud Transformers (Wang et al., 2025b). Prompting has also proven beneficial for multi-sensor 3D perception. In camera–LiDAR fusion, lightweight prompters can inject LiDAR-aware cues into camera-based 3D detectors to enhance depth reasoning. This technique adds negligible computational overhead and can be deployed at test time, even in the absence of LiDAR data (Guo & Ling, 2025). Modern visual–LiDAR 3D detection frameworks now employ soft prompts, inserted at various fusion stages, to guide cross-modal attention, an approach shown to achieve greater data efficiency on benchmarks like nuScenes (Li et al., 2025d). Prompt-based designs are also being explored for open-world retrieval and recognition tasks involving 3D queries, where negative prompts and complementary prompt heads have been shown to improve robustness against distribution shifts and unseen categories (Xu et al., 2024b).

Across these diverse studies, a clear pattern emerges: prompts serve as a lightweight yet controllable interface to steer large 2D or 3D backbones toward specialized point-cloud tasks. Input-space prompts are ideal for scenarios that involve repurposing existing image-based pretraining or require a model-agnostic solution. Conversely, token-level prompts are better suited for native 3D Transformers, where they can be injected as a few learnable tokens on a per-layer or per-instance basis. Geometry-aware and instance-adaptive variants achieve performance nearly on par with full fine-tuning while substantially reducing memory requirements. Finally, fusion-oriented prompting enables the seamless integration of multi-modal data, such as LiDAR and RGB, without necessitating backbone retraining.

The application of PA in 3D vision reveals a clear dichotomy based on the architecture of the pre-trained backbone. VP, often implemented via multi-view rendering or point-to-pixel projection, serves as a crucial modality bridge (Wang et al., 2022d; Zhu et al., 2023). Its primary utility lies in formatting sparse, irregular point clouds into dense, image-like inputs, enabling the direct reuse of powerful 2D vision-language models (*e.g.*, CLIP) without the need for expensive 3D pre-training. In contrast, VPT is the method of choice for adapting native 3D Transformers (Zha et al., 2023; Ai et al., 2025). Because 3D encoders must process permutation-invariant sets, injecting learnable tokens into the self-attention mechanism allows for the precise capture of local geometric structures and surface normals while maintaining high parameter efficiency (which is often lost in projection-based VP approaches).

### 4.7 Video Understanding and Temporal Perception

Video understanding tasks introduce a temporal dimension that complicates standard recognition, presenting challenges such as motion blur, frame-rate variability, and the need to model long-range dependencies. PA offers a lightweight mechanism for integrating temporal reasoning into frozen backbones, thereby obviating the need for full fine-tuning. A primary line of research involves augmenting video Transformers with *spatio–temporal prompt tokens.* Space–Time Prompting, for example, inserts prompts at selected layers and learns them end-to-end for class-incremental action recognition, which circumvents catastrophic forgetting while keeping the backbone frozen (Pei et al., 2023). Attention Prompt Tuning steers the self-attention mechanism using a small set of learnable prompts to improve action recognition with modest computational overhead (Bandara & Patel, 2024). To enhance feature stability against noise and camera motion, the Spatio-Temporal Prompting Network employs prompts that are shared across frames within a clip, thereby improving the robustness of the extracted video features (Sun et al., 2024a). Another practical approach operates within the compressed domain: Compressed Video Prompt Tuning learns prompts directly from motion vectors and residuals, a method that reduces decoding and training costs while maintaining competitive accuracy (Li et al., 2023a; 2025b).

Prompting is also utilized to adapt pretrained image–text models for video-centric tasks. For example, *Vita-CLIP* learns prompts for both the vision and text encoders of CLIP, enabling a single frozen back-

bone to balance performance between supervised and zero-shot action recognition (Wasim et al., 2023). Similarly, *TC-CLIP* injects temporally contextualized prompts that summarize clip dynamics, thereby producing stronger video-language alignment for retrieval and recognition tasks. Beyond static prompts, several methods dynamically generate or update prompts on a per-clip basis. STOP integrates spatial–temporal *dynamic* prompting, wherein a lightweight module predicts prompts conditioned on the input video, improving open-domain video understanding with a frozen backbone (Liu et al., 2025d). TP-CLIP investigates whether simple *temporal prompting* is sufficient for CLIP-based video recognition, demonstrating that clip-level temporal prompts can close a significant portion of the performance gap compared to more parameter-heavy adapters. Finally, PA methods have also proven effective in test-time and low-label regimes. *DTS-TPT* performs test-time prompt tuning with dual temporal synchronization to adapt to distribution shifts without requiring labels or gradient access to the backbone (Yan et al., 2024). Subsequent work extends this concept by creating a small support set dynamically and updating prompts at inference time to facilitate zero-shot video classification (Yan et al., 2025). Collectively, these results suggest an effective strategy: augmenting a frozen encoder with a few learned or generated spatio-temporal prompts that are specifically designed to target the temporal cues absent in the original model. This approach retains computational efficiency while addressing the key failure modes inherent to adapting static image models for video.

Adapting vision models to video tasks fundamentally requires bridging the gap between static spatial representation and dynamic temporal reasoning. In this context, VPT has established itself as the standard paradigm (Pei et al., 2023). Since the most powerful vision backbones are pre-trained on static images (*e.g.*, ImageNet), they inherently lack temporal awareness. Injecting learnable prompt tokens—specifically designed to encode time or motion—provides a parameter-efficient way to expand the model's attention mechanism from 2D spatial correlation to 3D spatio-temporal modeling, effectively "upgrading" the architecture without the massive cost of video pre-training. While VP appears in specialized niches like compressed-domain adaptation (Li et al., 2023a), token-level injection remains the most effective method for capturing long-range temporal dependencies and action dynamics.

## 4.8 Underwater and Adverse Environments

Underwater scenes and adverse weather conditions (*e.g.*, fog, rain, snow, low light) introduce severe domain shifts that challenge models pretrained on standard, clear-weather datasets. The underlying physics of these environments is the primary cause of degradation. In underwater settings, wavelength-dependent light attenuation and scattering cause color distortion (typically a blue-green shift) and reduce contrast, obscuring object boundaries and fine textures. Similarly, adverse weather introduces complex, non-linear corruptions: fog and haze reduce global contrast, rain produces reflective streaks and occlusions, and snow can drastically alter scene geometry and color distributions. In response, Prompt-based Adaptation (PA) has emerged as a powerful and practical methodology for enhancing model robustness in these specialized domains, often without requiring costly full-model retraining.

One prominent strategy involves adapting large-scale, promptable segmentation models like SAM through pixel-space visual prompts (VP). This approach is particularly effective because SAM's architecture is inherently designed to condition its output on spatial cues. By injecting prompts and fine-tuning lightweight adapters, these methods steer the frozen backbone toward domain-specific features. *WaterSAM*, for example, attaches low-rank adapters (LoRA) to SAM and uses traditional box or point prompts to segment organisms and objects in challenging underwater imagery (Hong et al., 2024). Taking this further, *USIS-SAM* develops a salient-feature prompter that automatically generates instance-specific cues, proving that intelligent prompt design significantly enhances recall in turbid water (Lian et al., 2024). A more recent pipeline, *UWSAM*, integrates an end-to-end underwater prompt generator to automatically synthesize prompts for diverse categories, demonstrating strong performance on its curated UIIS10K dataset (Li et al., 2025c). This paradigm extends effectively to terrestrial challenges as well. For instance, *SAM-EDA* uses semantic prompts to guide SAM for road segmentation in adverse weather, employing a teacher–assistant distillation scheme to transfer the performance gains into a compact student model (Wang et al., 2024c). The key advantage of these systems is their modularity; they preserve the fixed model internals and are compatible across different backbones, a crucial feature when only an API or a frozen checkpoint is accessible. When full model gradients are available, injecting learnable token prompts directly into the network (VPT) offers

a more potent mechanism for managing distribution shifts. These token prompts act as learnable "instructions" prepended to the input sequence of a Vision Transformer, conditioning the self-attention layers to focus on robust, domain-invariant features while ignoring nuisance variables. *DiffPrompter* introduces differentiable implicit prompts that are learned end-to-end for road segmentation, demonstrating resilience in adverse weather (Kalwar et al., 2023). Similarly, *CoDA* proposes a severity-aware visual prompt tuning mechanism for unsupervised domain adaptation. It groups scenes by difficulty (*e.g.*, light vs. heavy fog) and trains distinct prompt branches for low- and high-severity images. Although the prompts are discarded at inference, their influence is baked into the adapted model parameters, improving its generalization. For underwater semantic segmentation, *SEA-Net* applies VPT with severity perception and cross-domain priors to enhance performance across different sites and turbidity levels (He et al., 2025b). Orthogonal to severity cues, frequency-space information can be integrated with token prompts. *VFPT*, for example, uses Fourier components to guide features toward stable spectra, thereby improving robustness against weather-related corruptions with a negligible parameter increase (Zeng et al., 2024b). In summary, both underwater and adverse weather applications demand models with (i) strong geometry and shape priors to reconstruct boundaries degraded by scattering and noise, and (ii) explicit mechanisms to control for environmental nuisance factors like turbidity, haze, and illumination. Pixel-space prompting (VP) offers a straightforward, black-box-friendly solution that pairs effectively with SAM-style interfaces and PEFT techniques like LoRA. In contrast, token-level VPT affords finer-grained, white-box control over the model's internal representations when end-to-end training is feasible. A promising future direction lies in hybrid approaches that combine the strengths of both: an instance-adaptive generator could provide initial spatial prompts (VP), while a set of learned, condition-aware tokens (VPT) could simultaneously steer the model's feature extraction process to counteract the specific type and severity of environmental degradation.

Navigating the optical challenges of underwater and adverse weather environments requires addressing two distinct forms of degradation: spatial ambiguity and feature distortion. VP excels at the former, serving as a spatial anchor (Hong et al., 2024; Lian et al., 2024). In turbid or hazy conditions where object boundaries dissolve, pixel-level prompts (whether manual points or generated masks) provide the necessary localization cues to guide segmentation models like SAM, effectively compensating for the loss of high-frequency edge information. VPT, however, addresses the latter by enforcing feature invariance (Kalwar et al., 2023). By tuning internal tokens, these methods recalibrate the backbone's attention to suppress environmental nuisance factors (*e.g.*, the "blue-green" shift of water or the white noise of fog), ensuring that semantic representations remain robust even when the global distribution shifts drastically from the clear-weather pre-training domain.

## 5 PA under Practical Constraints

In the previous discussions, we focus on PA attempts on conventional supervised finetuning (§2-4). Currently, PA successfully demonstrates remarkable effectiveness in a variety of learning scenarios defined by significant operational constraints Oh et al. (2023); Zhang et al. (2025e;d); Yu et al. (2023); Khattak et al. (2023). The reason is due to its parameter-efficient and data-efficient nature, making it an ideal candidate for situations where data is scarce, data distributions are non-stationary Zhang et al. (2025e); Wang et al. (2022c); Zhao et al. (2024a), or access to model internals Oh et al. (2023); Tsai et al. (2020) and computational resources is limited Zhang et al. (2025e;d); Niu et al. (2024); Zhao et al. (2025a). In this section, we categorize and survey the application of PA across these challenging paradigms. Formally, we organize these paradigms into three core areas: adaptation under data constraints (see §5.1), adaptation in dynamic environments (see §5.2), and adaptation with resource or access limitations (see §5.3).

### 5.1 Data-Constrained Adaptation

Data-constrained adaptation includes scenarios where the primary limitation is the quantity or quality of labeled data. Under this condition, the proposed methods should maximize their learning effectiveness from minimal supervision or even entirely unlabeled datasets.

Table 3: **Comprehensive summary of VPT methods across domains.**

| Domain | Method | Task | Sub-type |
|---|---|---|---|
| Medical | ProSFDA (Hu et al., 2022) | Source-free domain adaptation with prompt tuning | VPT-Learnable |
| Medical | Biomed-DPT (Peng et al., 2025) | Dual-modality visual prompt for medical reasoning | VPT-Learnable |
| Medical | FedLPPA (Lin et al., 2024a) | Federated personalized prompt for segmentation | VPT-Learnable |
| RS | De-Prompting (Liu et al., 2023) | Temporal prompt tuning for change captioning | VPT-Learnable |
| RS | VPT-CLIP (Liu et al., 2025c) | Bi-temporal change detection via CLIP priors | VPT-Learnable |
| RS | PEFTT (Yuan et al., 2023) | Prompt-based lightweight satellite–caption retrieval | VPT-Learnable |
| RS | RLita (Zhang et al., 2025b) | Prompt-tuned cross-modal retrieval in RS | VPT-Learnable |
| RS | MVP (Zhu et al., 2024a) | Few-shot classification with spatial prior prompts | VPT-Learnable |
| RS | UPETU (Dong et al., 2024b) | Universal PEFT encoder for RS backbones | VPT-Learnable |
| RS | LayerLink (Zhu et al., 2025) | Layer-wise linking for efficient prompt adaptation | VPT-Learnable |
| 3D | PointCLIP V2 (Zhu et al., 2023) | CLIP-to-point cloud alignment via token prompts | VPT-Learnable |
| 3D | CLIP2Point (Huang et al., 2023c) | Image–point cloud alignment using token prompts | VPT-Learnable |
| 3D | IDPT (Zha et al., 2023) | Instance-aware dynamic token prompts | VPT-Generated |
| 3D | GAPrompt (Ai et al., 2025) | Geometry prompts for local and global structure | VPT-Learnable |
| 3D | PosPrompt3D (Zhang et al., 2024c) | Positional prompt tuning for 3D transformers | VPT-Learnable |
| 3D | Point-PRC (Sun et al., 2024b) | Regularization between task and general prompts | VPT-Learnable |
| 3D | PointLoRA (Wang et al., 2025b) | Low-rank adapters + prompts for 3D Transformers | VPT-Learnable |
| 3D | PromptDet (Guo & Ling, 2025) | Camera–LiDAR fusion with soft prompts | VPT-Learnable |
| 3D | PF3Det (Li et al., 2025d) | Multi-stage LiDAR–camera fusion with prompts | VPT-Learnable |
| 3D | NPCP (Xu et al., 2024b) | Negative prompts for robust 3D retrieval | VPT-Learnable |
| Autonomous | UniUVPT (Ma et al., 2023) | Source-free domain adaptation for driving datasets | VPT-Learnable |
| Autonomous | BEVCLIP (Xie et al., 2025b) | Shared prompts for BEV retrieval/segmentation | VPT-Learnable |
| Autonomous | BEVInstructor (Yang et al., 2024c) | Perspective-to-BEV generation via prompt | VPT-Learnable |
| Video | ST-Prompting (Pei et al., 2023) | Spatio-temporal token prompts for video recognition | VPT-Learnable |
| Video | APT (Bandara & Patel, 2024) | Attention steering with learnable prompts | VPT-Learnable |
| Video | STPN (Sun et al., 2024a) | Shared prompts across frames for video features | VPT-Learnable |
| Video | CV-PT (Li et al., 2023a) | Prompts learned from motion vectors/residuals | VPT-Learnable |
| Video | Vita-CLIP (Wasim et al., 2023) | Dual (vision/text) prompts for zero-shot recognition | VPT-Learnable |
| Video | STOP (Liu et al., 2025d) | Dynamic generator producing per-clip prompts | VPT-Generated |
| Video | DTS-TPT (Yan et al., 2024) | Test-time dual-synchronization prompt tuning | VPT-Learnable |
| Video | TESTV (Yan et al., 2025) | Dynamic support set and prompt update at inference | VPT-Learnable |
| Underwater | SEA-Net (He et al., 2025b) | Severity-aware VPT for underwater segmentation | VPT-Learnable |

### 5.1.1 Few-Shot Learning

In the few-shot learning (FSL) setting, a model must generalize from a very small number of labeled examples. Fully fine-tuning a large-scale model in FSL risks severe overfitting, while VPT can potentially preserve the robust, generalizable features of the frozen backbone. A key challenge is the "Base-New Trade-off," where optimizing for seen (base) classes degrades performance on unseen (new) classes. Advanced methods address this by creating more dynamic and context-aware prompts. For instance, *MaPLe* (Khattak et al., 2023) introduces multi-modal prompt learning, creating learnable prompts in both the vision and language encoders of a Vision-Language Model (VLM). Crucially, it uses a "coupling function" to link the vision and language prompts, ensuring they are optimized synergistically to improve cross-modal alignment and enhance generalization from limited data.

### 5.1.2 Unsupervised Domain Adaptation

Unsupervised Domain Adaptation (UDA) aims to adapt a model trained on a labeled source domain to an unlabeled target domain. While many modern techniques are presented in the context of Test-Time Adaptation (TTA), their core mechanisms are directly applicable to the offline UDA setting. In this context,

prompts must be optimized without direct supervision. For example, the *DePT* (Gao et al., 2022) tunes only source-initialized prompts at test time, using online pseudo-labeling and a hierarchical self-supervised regularization to adapt efficiently, even with very limited target data. Similarly, *OT-VP* (Zhang et al., 2025d) provides a principled solution by framing UDA as a distribution alignment problem. It learns a universal visual prompt for the target domain by minimizing the Optimal Transport (OT) distance between the feature distributions of the prompted target data and the source data. These approaches highlight how VPT can effectively bridge the domain gap in unsupervised settings.

### 5.1.3 Multimodal Learning with Missing Modalities

A significant real-world challenge is handling multimodal data where one or more modalities (*e.g.*, text accompanying an image) may be missing during training or inference. PA offers an effective, parameter-efficient solution. (Lee et al., 2023) introduce modality-missing-aware prompts, where different learnable prompts are assigned to different missing-modality cases (*e.g.*, image-only, text-only, complete). This allows a frozen multimodal transformer to adapt its behavior based on the available modalities. Building on this, *DPL* (Lu et al., 2025) proposes a decoupled prototype-based output head that can be integrated with prompt-based methods. This work uses missing-case-aware, class-wise prototypes for each modality, further improving the model's robustness by specializing the classification head itself to the missing modality scenario.

## 5.2 Dynamic Adaptation

Dynamic adaptation refers to scenarios in which the data distribution evolves over time, necessitating continuous and efficient model adjustment.

### 5.2.1 Test-Time Adaptation

Test-Time Adaptation (TTA) involves adapting a model to a new target domain during inference, often in an online setting where data arrives sequentially. The efficiency of VPT is highly suitable for this paradigm. *DynaPrompt* (Xiao et al., 2025e) introduces dynamic test-time prompt tuning, which generates input-dependent prompts for more precise adaptation. For scenarios where the distribution continuously shifts, *DPCore* (Zhang et al., 2025e) maintains a coreset of dynamic prompts to efficiently adapt to evolving domains without catastrophic forgetting. Furthermore, some methods operate under even stricter constraints; *FOA* (Niu et al., 2024) developed a TTA method requiring only forward passes, making it extremely efficient. In the zero-shot setting, *PromptAlign* (Samadh et al., 2023) uses distribution alignment to adapt prompts at test time for generalization to unseen classes and domains. Finally, tackling the challenge where models are only accessible via APIs without gradients, Zhang et al. (2026) propose a stable black-box TTA framework. Their method circumvents the instability and high query costs of zeroth-order optimization by leveraging visual prompting as the core mechanism, using a locally updated surrogate to optimize the prompt and effectively steer the black-box model.

### 5.2.2 Continual and Incremental Learning

Continual Learning (CL) (Mai et al., 2022) involves training on a sequence of tasks without forgetting prior ones. VPT-based approaches reframe this from a weight-regularization problem to a prompt management problem. The seminal work *L2P* (Wang et al., 2022c) introduces a "prompt pool" that acts as a key-value memory. For any input, the model queries the pool to select the most relevant prompts to prepend to the input sequence, allowing it to dynamically compose "instructions" for the frozen backbone. Building on this, *DualPrompt* (Wang et al., 2022b) refines the approach by drawing inspiration from Complementary Learning Systems (CLS) theory. Instead of a single pool, it explicitly decouples knowledge into two sets of prompts: a shared G-Prompt (General) to learn task-invariant knowledge and a set of E-Prompts (Expert) to capture task-specific knowledge. This design more effectively isolates knowledge for dissimilar tasks while sharing common knowledge, leading to further reductions in catastrophic forgetting.

### 5.3 Resource and Access-Constrained Adaptation

Resource and access-constrained adaptation focuses on limitations imposed by model accessibility and the computational environment, such as black-box APIs and decentralized data settings.

#### 5.3.1 Black-Box and Gray-Box Model Adaptation

In a black-box setting, a model is only accessible via an API, with no access to its internal parameters or gradients, making standard prompt tuning impossible. To solve this, methods have turned to gradient-free optimization. *BlackVIP* (Oh et al., 2023) pioneers this by using a small generator network (*i.e.*, a "Coordinator") to create an input-dependent visual prompt. The parameters of this tiny generator are then optimized using a zeroth-order algorithm that estimates gradients by making a small number of queries to the black-box model. This builds on earlier ideas of model reprogramming, which showed that black-box models could be repurposed for new tasks with scarce data (Tsai et al., 2020). For Vision-Language Models offered as a service, other black-box prompt tuning methods have been developed to optimize prompts in a derivative-free manner (Yu et al., 2023). However, these zeroth-order approaches can be query-intensive and unstable. To mitigate this, Zhang et al. (2025c) proposes a prime-then-reprogram strategy that shifts the optimization burden to a local module, significantly reducing API costs. In the context of test-time adaptation, Zhang et al. (2026) further addresses optimization instability by introducing a framework that steers the black-box model using a locally updated surrogate.

#### 5.3.2 Federated and Decentralized Learning

In Federated Learning (FL), data is distributed across clients and cannot be centralized, posing challenges of communication overhead and statistical heterogeneity (*i.e.*, non-IID data). VPT is a natural fit, as it drastically reduces communication costs by requiring clients to transmit only small prompt parameters. To handle non-IID data, personalized FL approaches are emerging, as demonstrated in works like *pFedPrompt* (Guo et al., 2023) and *FedPrompt* (Zhao et al., 2023). In this paradigm, clients learn a combination of shared global prompts, which are aggregated at a server to capture collective knowledge, and private local prompts, which are trained only on the client's data to capture its unique distribution. This concept of decoupling prompts, central to *pFedPrompt*, allows for personalized models that benefit from collaboration while being tailored to local data characteristics.

## 6 Trustworthy AI

PA is increasingly used as a lightweight alternative to full fine-tuning. In trustworthy AI, PA mainly supports three goals: robustness (see §6.1), fairness and bias mitigation (see §6.2), and privacy and security (see §6.3). Below, we review representative designs and summarize practical takeaways for deployment.

### 6.1 Robustness

Robustness is a core requirement for trustworthy AI: models should be reliable under distribution shifts and resist adversarial manipulations. Large pre-trained models still degrade in these cases. PA on pixels (VP) or internal tokens (VPT) offers a low-cost handle to improve robustness without touching the backbone. We discuss two threads: *domain shift* and *adversarial robustness*.

**Domain shift.** VPT is a natural tool for cross-domain transfer. According to target access, methods fall into domain generalization (DG; no target data) and unsupervised domain adaptation (UDA; unlabeled target seen in training).

For DG, (Li et al., 2022a) introduce common–specific prompts to capture domain-shared and sample-specific cues. (Bai et al., 2024b) propose a soft prompt generator that emits instance-specific prompts conditioned on domain information. *EPVT* (Yan et al., 2023) uses a low-rank prompt generator to reduce artifact bias in lesion recognition. *DAPSAM* (Wei et al., 2024) adapts SAM via domain-adaptive prompts from prototype memories, improving cross-domain performance. On the VL side but with **visual** prompts, (Cheng

et al., 2024) disentangle text-guided visual prompts into domain-invariant and domain-specific parts; *ODG-CLIP* (Singha et al., 2024) attaches specialized prompts for unknown classes. *StyLIP* (Bose et al., 2024) separates style and content across scales; (Gupta et al., 2025) use cross-attentive prompts to mix domain- and class-generic tokens.

For UDA, *USDAP* (Shao et al., 2024) learns prompts to align target and source distributions. (Zhan et al., 2024) improves robustness to noisy pseudo labels with dynamic mask prompting. In adversarial UDA, (Jin et al., 2023) meta-optimizes prompts to relabel target samples adaptively; *ADAPT* (Cui et al., 2025) employs visual prompts in a minimax game to align both global and category distributions. In medical segmentation, *ProSFDA* (Hu et al., 2022) and *DDFP* (Yin et al., 2025) reduce cross-domain gaps with prompt-based styles or frequency cues.

**Adversarial robustness.** Beyond natural shift, attacks craft imperceptible perturbations to force errors. Adversarial training is strong but costly; PA gives a cheaper alternative.

APD (Luo et al., 2024) learns prompts that align perturbed features with robust embeddings, improving the robustness–accuracy trade-off. *AMPT* (Zhao et al., 2025b) uses a pool of prompts and combines them dynamically. (Zhou et al., 2024c) improves few-shot robustness by enforcing consistency between clean and adversarial features. *ER-APT* (Jia et al., 2025) optimizes region-level prompts via evolutionary search to resist diverse attacks.

Test-time and structural prompting are also explored. *RobustMAE* (Huang et al., 2023a) inserts frequency-domain prompts to occupy high-frequency bands at inference. *PBL* (Li et al., 2023b) transfers robustness from a robust source by loosening decision boundaries. *ARVP* (Liu & Li, 2024) applies adversarial reprogramming with visual prompts in class-incremental learning to reduce forgetting. Depth-wise adversarial prompt tuning *(MDAPT)* (Li et al., 2025a) injects prompts across layers; *TAPT* (Wang et al., 2025d) adapts prompts per sample to secure zero-shot inference under strong attacks.

PA methods reduce trainable state and are easy to add on top of frozen encoders. They help when labels are scarce or compute is tight. For large activation footprints, pair PA with memory-efficient training (cf. §2.4); for open-world shifts, prefer instance-adaptive or frequency-aware prompts.

## 6.2   Fairness and Bias Mitigation

Prompts have also been used to mitigate social biases, although most methods target vision-language models (VLMs) and focus on the text branch or cross-modal alignment rather than the PA within the vision backbone that is the focus of this survey. Representative approaches fall into three main categories.

**(1) Debiasing via Subspace Projection.**  *Biased Prompts* (Chuang et al., 2023) define a bias direction in the embedding space using a set of text prompts that describe the bias. By projecting this direction out of the feature space, it improves the fairness and robustness of CLIP without requiring retraining. This approach modifies only the text embeddings but is effective for both discriminative and generative models.

**(2) Adversarial or Unified Debiasing with Prompts.**   (Berg et al., 2022) reduce gender and skin-tone biases in CLIP by prepending a small number of learnable prompts to text queries and employing adversarial training. This approach requires minimal computation and no access to the original training data. More recently, *SFID* (Jung et al., 2024) introduced a unified debiasing framework that reduces multimodal biases via feature clipping and confidence patching without degrading vision–language alignment.

**(3) Joint Alignment and Debiasing of Image and Text Modalities.**   Recent work indicates that modifying only the text branch can degrade cross-modal alignment. To address this, (Zhang et al., 2025a) propose a method to jointly align and debias both the image and text modalities, mitigating the trade-off between performance and fairness. Other analytical studies also suggest that the vision branch is often a primary source of bias and that fairness across client groups must be considered in federated learning settings (Weng et al., 2024; Wang et al., 2025e).

**Relation to the Scope of This Survey.** The methods above are primarily examples of VLM prompting or text-side correction. They provide evidence that prompts can serve as a control plane for debiasing, but they are not direct applications of PA via pixel (VP) or token injection (VPT) into the vision backbone. We reference them in this survey to: (i) provide a contrasting perspective, and (ii) highlight that by situating learnable prompts within the vision backbone (VPT) or in the pixel space (VP), debiasing and domain generalization can be implemented as parameter-efficient, modular components.

### 6.3 Privacy and Security

PA is also implicated in model security, particularly concerning backdoor attacks and copyright protection. While research in this area again focuses primarily on VLMs, the methodologies offer valuable insights into the threat surface and potential defenses for PA.

**(1) Backdoor Attacks during the Prompt Learning Phase.** *BadCLIP* (Bai et al., 2024a) demonstrates that backdoors can be implanted into CLIP during the prompt learning stage. It jointly optimizes a learnable image trigger with a trigger-aware text prompt generator to achieve a high attack success rate while preserving accuracy on clean samples. This directly shows that learnable or generated prompts are themselves part of the attack surface. More broadly, recent work has systematically evaluated and extended backdoor attacks on LVLMs, including vulnerabilities in cross-domain and instruction-tuning scenarios (Liang et al., 2025), as well as training-data-free, test-time backdoors like *AnyDoor* (Lu et al., 2024). These findings suggest that even with a frozen backbone, adaptation that relies solely on prompts requires corresponding security audits.

**(2) Prompt-based Detection and Defense.** One line of defense uses prompt tuning to detect backdoored samples, for instance, by discriminating trigger consistency or separability (Stein et al., 2024). Another approach introduces mechanisms like repulsive activation at the representation level for unified defense. Although mostly validated on VLMs, these ideas can be transferred to VPT, for example by using a small discriminator head or generator to evaluate anomalous coupling between prompts and representations.

**(3) Prompt Copyright and Watermarking.** *WVPrompt* (Ren et al., 2024) treats a watermark as a backdoor injected into prompts, enabling remote copyright verification through statistical tests. This work highlights the security property of *prompts as assets*. For VPT systems that use generators to produce prompts, such watermarks could also serve as a mechanism for verifying usage compliance and tracking model versions.

## 7 Foundational Analysis and Theory of PA

Understanding the theoretical foundations of PA is essential beyond its demonstrated practical advantages. However, we should acknowledge that the related theoretical analysis remains limited in the current community. In this section, we aim to explore some of the fundamental questions, where some questions are general (*i.e.*, Q1—3), some are specified for VP or VPT (*i.e.*, Q4, Q5).

- *General Q1: How does PA induce behavioral changes in the model?*

  For both VP and VPT paradigms, gradient-weighted class activation mapping (GradCAM) (Selvaraju et al., 2017; Chakraborty et al., 2022; Zeng et al., 2024b) offers an intuitive visual explanation of PA's decision-making process.

  For VP, (Rezaei et al., 2024) shows that learned prompts can explicitly steer the attention of vision models with GradCAM visualizations. In particular, their experiments reveal that the added prompt pixels bias the model's attention maps toward the spatial location of the prompts, thereby altering which regions of the image the model emphasizes during decision-making. This suggests that VP modifies early feature activations in a way that reallocates attention, providing a concrete mechanism by which

the prompt drives behavioral change in the frozen model.

For VPT, (Han et al., 2024) uses the attention visualizations to reveal behavioral differences between VPT and full fine-tuning. The GradCAM analysis highlights instances where full fine-tuning fails to recognize objects, whereas prompt tuning achieves correct classifications. In these successful cases, distinct visual explanation patterns emerge through heatmaps. For instance, in an image of a bicycle where full fine-tuning fails, prompt tuning can identify the bicycle by attending to its structural features.

While the research on PA's influence on model behaviors remains limited (*i.e.*, partially due to PA's natural incompatibility with the linear representation hypothesis (Liu et al., 2025b; Wu et al., 2024b; Geiger et al., 2021) or other interpretability approaches such as concept-based analysis (Parekh et al., 2024) or causal probing (Geiger et al., 2021; 2024) , current approaches suggest that PA can effectively direct the model's attention to salient image regions, enhancing the overall performance.

- *General Q2: What do visual prompts learn?*

  In learnable VP, (Chen et al., 2023) shows that visual prompts primarily learn to bridge the gap between the pretrained model's label space and the downstream target classes. Their analysis demonstrates that the quality of this implicit label mapping (*i.e.*, how well source and target categories align) directly governs VP performance. In other words, VP does not endow the model with new semantics but instead learns pixel patterns that re-map pretrained representations to new class labels, effectively repurposing existing knowledge through input-space adaptation.

  In learnable VPT, recent analysis of the training dynamics reveals that prompt tokens exhibit a specific learning pattern during fine-tuning. (Wang et al., 2024b) conducts an empirical study measuring the normalized mutual information (NMI) (Estévez et al., 2009) between prompt tokens and patch tokens across different transformer layers during training. Specifically, NMI is computed using sigmoid-normalized cross-attention between prompts and patch tokens, approximating the joint distribution as:

  $$\text{NMI}(P_i; E_i) = \frac{2 \times I(P_i; E_i)}{H(P_i) + H(E_i)}, \tag{10}$$

  where $P_i$ and $E_i$ represent the prompt tokens and patch tokens at the $(i)$-th transformer layer, respectively, $I$ denotes the standard mutual information, and $H(\cdot)$ represents the entropy. Their empirical observation across four datasets (*i.e.*, CUB-200-2011, Caltech-101, Patch Camelyon, and Clevrcount) shows that visual prompts learn representations that increasingly align with the patch token distributions throughout the training process (*i.e.*, distribution of prompts for downstream contextualization gradually converges towards the distribution of patch tokens), suggesting that effective prompt learning involves establishing stronger correlations with image patch embeddings rather than learning completely independent representations.

- *General Q3: How effective are PA methods across different adaptation settings?*

  The effectiveness of PA approaches varies with the nature of the adaptation setting. Recent studies reveal that while VP demonstrate notable robustness to distribution shifts, VPT's performance is more sensitive to the relationship between source and target tasks and data distributions.

  For VP, (Sun et al., 2024c) demonstrates that VP can exhibit notable robustness against distribution shift, highlighting its effectiveness in handling out-of-distribution (OOD) settings. Empirical evaluations on benchmarks such as WILDS show that VP achieves consistent gains compared to conventional prompting strategies and even outperforms strong baselines like linear probing and full fine-tuning in certain cases. This robustness extends beyond domain shifts: when tested on corruption datasets such as CIFAR100-C and CIFAR10-C, VP maintains strong performance under diverse perturbations, rivaling the results of full fine-tuned models. These findings suggest that VP's standalone design provides better resilience to data variations. This property makes VP a compelling choice for scenarios where models

are expected to generalize under distribution shifts or noisy conditions.

For VPT, (Han et al., 2024; Mai et al., 2025) presents a comprehensive empirical study across 19 visual tasks in the VTAB-1k benchmark, identifying key conditions under which VPT yields superior performance. The authors show that the relative effectiveness of VPT *vs*. FT depends critically on two factors: the similarity of data distributions between the pretraining and downstream tasks, and the disparity in task objectives. Specifically, VPT tends to outperform FT when (1) the downstream task is substantially different from the pretraining objective (*e.g.*, classification *vs*. spatial reasoning tasks like counting or distance estimation), or (2) the data distributions between the source and target domains are closely aligned (*e.g.*, natural images in both cases). Analyses along these two axes often reveal that VPT is well-suited to three out of the four transfer learning scenarios defined by combinations of task and data similarity. When either the task objectives differ substantially from those seen during pretraining or the data distributions are closely aligned, VPT tends to provide better performance under low-resource settings. This is attributed to its ability to preserve the pretrained model's representations while introducing only a small set of task-specific parameters. In such cases, VPT offers a favorable trade-off between adaptation capacity and parameter efficiency. However, as the amount of downstream data increases, the performance advantage of VPT diminishes, and full finetuning may become preferable. These findings suggest that the choice of tuning strategy, particularly the prompt length and extent of parameter updates, should be informed by both the task formulation and the distributional relationship between the source and target domains.

- ***VP Q4: What is the best way to place the prompt in VP?***

  Prompt placement plays a critical role in how VP methods interact with pretrained vision models, and the optimal strategy differs notably between fixed and learnable VP variants.

  For fixed VP methods, primarily developed for segmentation-oriented tasks, prompts are directly plotted or overlaid on the input image. A representative example is SAM (Kirillov et al., 2023), where prompts such as points, boxes, or masks are spatially anchored to image regions to guide the model's attention toward target objects or segments. These spatially explicit prompts serve as conditioning cues rather than learnable parameters, making their placement inherently task-driven and interpretable. In such frameworks, the prompt's position corresponds directly to the semantic location of the object or region of interest, effectively functioning as spatial supervision for dense prediction tasks.

  In contrast, learnable VP methods adopt a padding-based placement strategy to preserve the integrity of visual content while optimizing prompt effectiveness. (Sun et al., 2024c) demonstrates that the most effective way to place visual prompts is by padding learnable pixels around the image rather than embedding them within it. Their design slightly shrinks the original image and surrounds it with a non-overlapping frame of prompt pixels, ensuring that the visual content remains intact while the prompt interacts with the model's spatial representations through positional embeddings. This "border prompt" approach prevents corruption of salient image regions, stabilizes optimization, and consistently outperforms additive or internal prompt placements across diverse datasets. While other studies, such as LoR-VP (Jin et al., 2025) and SVDP (Yang et al., 2024a), have explored interior or adaptive prompt placements for dense prediction and domain adaptation, the border-padding strategy remains the most reliable and empirically validated prompt placement for image classification with frozen vision backbones.

- ***VPT Q5: What is the optimal prompt length for VPT?***

  Determining the optimal prompt length in VPT is crucial for balancing model performance and computational efficiency. Empirical findings by (Kim et al., 2024b) have shown that the relationship between prompt quantity and fine-tuning accuracy is non-linear, refuting the assumption that more prompts always lead to better performance. Notably, reducing prompt length can result in minimal accuracy loss, with most performance degradation occurring at lower prompt ranges. This is theoretically supported by the low-rank characteristics of self-attention matrices in Vision Transformers. The rank increase of

the self-attention matrix with added prompts follows a logarithmic trend:

$$\text{rank}(\tilde{A}_{n+m}) - \text{rank}(\tilde{A}_n) = O(\log(m)). \tag{11}$$

This explains the diminishing returns of adding more prompts, as initial prompts contribute more significantly to attention than later ones. From a computational standpoint, prompt length directly affects efficiency, with longer prompts introducing substantial overhead. Thus, the optimal prompt length should reflect a balance between accuracy and resource limits. Recent works (Han et al., 2023; Shang et al., 2025; Xiao et al., 2025c; **?**) also support the theoretical view that prompt quality does not inherently depend on prompt length. Instead, a small number of well-chosen prompts can effectively handle the downstream fine-tuning, emphasizing that informativeness and alignment might matter more than quantity. This observation aligns with language domain-related research (Zeng et al., 2025; Liu et al., 2025a).

## 8 Discussion and Challenges

**Guideline for Selection: VP vs. VPT.** A recurring theme throughout our survey is the trade-off between the flexibility of input-space prompting and the adaptation capacity of internal prompt tuning. Based on the empirical evidence reviewed across domains, we synthesize the following comparative guidelines:

- **Scenario 1: Black-box Access and Deployment Flexibility.** When model internals are inaccessible (*e.g.*, API-based foundation models) or when the deployment environment requires strict architectural invariance, VP is the superior choice (Oh et al., 2023). Pixel-level prompts serve as an external "plug-and-play" module that can be optimized via zeroth-order methods or generated by lightweight networks, effectively circumventing the need for gradient access. This makes VP particularly dominant in interactive segmentation (*e.g.*, SAM-based medical or industrial applications) (Kirillov et al., 2023; Chen et al., 2024a), where spatial cues (points/boxes) are more intuitive than semantic tokens.

- **Scenario 2: Deep Semantic Adaptation and Domain Shifts.** For tasks involving significant domain shifts (*e.g.*, natural images to remote sensing or medical reports) or requiring temporal/modality alignment (*e.g.*, video understanding), VPT generally yields better performance trade-offs (Han et al., 2024; Pei et al., 2023). Unlike VP, which is limited to modifying the input distribution, VPT injects learnable tokens into deep transformer layers, allowing for the direct recalibration of high-level semantic features. This capacity is essential for tasks where the pre-trained features are semantically misaligned with the downstream objective, such as in 3D point cloud classification (Zha et al., 2023) or long-tailed recognition (Dong et al., 2023).

- **Scenario 3: Efficiency Constraints.** While both methods are parameter-efficient, their memory footprints differ. If training memory is the primary bottleneck, Fixed VP or inference-only adapters should be prioritized as they bypass backbone backpropagation. However, if the goal is to maximize accuracy within a limited parameter budget (*i.e.*, $< 1\%$) while tolerating standard training memory costs, VPT Deep consistently outperforms shallow VP variants by leveraging the full expressivity of the deep backbone (Jia et al., 2022; Han et al., 2023).

In summary, the choice between VP and VPT should be driven by the *access level* (black-box vs. white-box) and the *adaptation depth* (spatial correction vs. semantic realignment) required by the target application.

Despite the significance of prompt-based adaptation methods, there are critical challenges to using it in practice. In the following, we identify some of them.

- **Safety Alignment.** Safety alignment is pivotal for the advancement of AI technologies. The safety of PA approaches should be woven into their development and deployment. Key elements in this process include interpretability, governance, and rigorous verification of model properties. As stated in §2, current PA approaches can be understood as a targeted intervention towards expected data distributions. Such interventions, however, can involve malicious actors, who may fine-tune models to generate or amplify

harmful content, misinformation, or biased outputs (see §6.3). To address these concerns, mitigation strategies can be considered. These include robustness evaluations, continuous monitoring of model behavior, and systematic bias audits. Additionally, comprehensive documentation of models and training datasets, alongside transparent disclosure of any known biases introduced during model development, is essential (Liu et al., 2025b).

In sum, aligning intervention directions with human values, goals, and expectations remains a critical challenge. Advancing this alignment requires sustained research efforts, particularly in detecting instances of misalignment and formulating corrective mechanisms.

- **Training Overhead and Stability.** While PA approaches offer enhanced training parameter efficiency, they present notable limitations during training. Due to the different characteristics of VP and VPT, we discuss them separately.

  For VP, certain approaches (Huang et al., 2023b), introduce task-specific or cluster-specific prompting. These approaches necessitate additional training data for clustering and increase both prompt parameter count and optimization complexity, thereby imposing substantial additional training overhead. VP also tends to exhibit instability: whether in design, location, or pattern, minor perturbations in prompt configuration can result in significant performance degradation. This sensitivity undermines its robustness and generalization across diverse tasks or datasets. Moreover, when VP is applied to robust source models, it often inherits a trade-off: adversarial robustness is maintained at the cost of noticeable declines in standard accuracy (Li et al., 2023b).

  For VPT, the first challenge lies in the training overhead. Although VPT reduces the per-iteration training time by updating a subset of the model parameters via gradient descent, the total training duration often increases significantly compared to full fine-tuning. This is primarily due to the need for extensive hyperparameter search across prompt length, learning rates, weight decay values, *etc.* A promising direction involves fixing certain hyperparameters to reduce the overall search space during training. Another key challenge arises from the instability of training outcomes under different initialization values (*i.e.*, random seeds). While VPT has demonstrated certain robustness across various initialization strategies (*e.g.*, *He* (He et al., 2015), *Truncated Normalization* (Paszke et al., 2019)), it exhibits greater variance (*i.e.*, more fluctuating results compared to full fine-tuning). This naturally necessitates additional training to achieve satisfactory results, thereby substantially diminishing the effectiveness of current efforts on reducing per-iteration training time. It is worth noticing that the training results of VPT are also influenced by different pre-trained strategies (*e.g.*, supervised objectives, self-supervised objectives: MAE (He et al., 2022) and MoCo v3 (Chen et al., 2021)).

  To sum up, reducing PA's training overhead and strengthening the robustness of training remain critical challenges. Intensified research efforts are needed, especially in developing training shortcuts, detecting, and developing strategies to rectify training stability.

- **Inference Latency.** PA often experiences increased inference latency due to the additional components supplemented to the input (*i.e.*, VP) or appended to the original vision models (*i.e.*, VPT). This add-on naturally introduces challenges related to additional memory consumption. In response, the research community has actively sought to alleviate such inference bottlenecks. Techniques to cut down on memory demands, both in terms of size and bandwidth, and to speed up inference computations have been devised. For example, pruning-based approaches have been developed to strategically remove certain less influential components from prompting. Other approaches, such as knowledge distillation, quantization, and the combination of memory-efficient fine-tuning (Simoulin et al., 2024; Choi et al., 2025), can also decrease memory usage and enhance computational throughput by lowering the precision of model weights and activations post-training.

- **Evaluation on Real-world Environments.** The selection of pre-trained models and the evaluation of PA, have predominantly relied on standardized academic benchmarks, such as VTAB-1k, FGVC, ImageNet. Such datasets, despite their significance in the evolution of AI, have limitations in their ability to reflect real-world characteristics accurately (*i.e.*, in other words, the robustness in distribution shifting). To truly confirm the capabilities and practical applicability of PA, it is imperative to assess them using data that is diverse, complex, and mirrors real-world scenarios. While this area remains

unexplored in VP, some research on VPT has shown that with measurable data distribution shifting, the results can be noticeably different (Han et al., 2024), revealing similarities to prompt tuning approaches in NLP (Wang et al., 2022a; Chen et al., 2022). Other studies on test-time adaptation (Xiao & Snoek, 2024; Lee et al., 2022; Huang et al., 2021; Li et al., 2023c) and learning (Banerjee et al., 2021; Ma et al., 2025; Xie et al., 2025a; Wang et al., 2025d; Mai et al., 2024) have provided initial evidence supporting the effectiveness of distributional calibration/correction techniques. Building on this observation, future work should prioritize robust methods for closing this gap. Addressing this challenge has the potential to significantly advance the landscape of PA, enhancing its adaptability across diverse visual contexts, including complex and heterogeneous scenarios encountered in AI for Science applications.

## 9 Conclusion

In this survey, we provide the first comprehensive review of prompt-based adaptation (PA) methods in large vision models. We highlight the fundamental differences between the two mainstream adaptation approaches: visual prompting (VP) and visual prompt tuning (VPT), and further discuss their applications, foundational analysis and theories, and current challenges. Given PA methods as lightweight and effective alternatives to full-fine-tuning under certain critical conditions, such as limited data, we hope our survey would enable researchers and practitioners to leverage their potential and drive innovation further in this area.

### Acknowledgments

This manuscript has been co-authored by ORNL, operated by UT-Battelle, LLC under Contract No. DE-AC05-00OR22725 with the U.S. Department of Energy. Any subjective views or opinions that might be expressed in the paper do not necessarily represent the views of the U.S. Department of Energy or the United States Government. This research was supported by the National Science Foundation under Grant No. 2450068. This work used NCSA Delta GPU through allocation CIS250460 from the Advanced Cyberinfrastructure Coordination Ecosystem: Services & Support (ACCESS) program, which is supported by U.S. National Science Foundation grants No. 2138259, No. 2138286, No. 2138307, No. 2137603, and No. 2138296.

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
