# OpenReview forum: "Prompt-based Adaptation in Large-scale Vision Models: A Survey"
_TMLR — Accepted by TMLR_

### Review · Reviewer_ajGX · 2025-12-06

**Summary Of Contributions:**

The manuscript provides a comprehensive survey of prompt-based adaptation (PA) techniques for large-scale vision models, with a clear focus on visual prompting (VP) and visual prompt tuning (VPT) in purely visual settings. In general, it could serve as a great resource for students and researchers. The survey motivates PA as a lightweight alternative to full-fine tuning and related methods, however, I think some quantitative results could further strengthen this position.

Strengths:
- Offers a clear taxonomy and articulate the distinction between VP and VPT to help reduce ambiguity in existing literature
- The survey spans a wide range of applications, from traditional computer vision tasks to domain-specific use cases, which is valuable for researchers exploring new directions
- Section 2.4’s detailed breakdown of memory footprint is particularly helpful, as such analysis is often missing in related work.
- The manuscript is well written and easy to follow


Weaknesses:
- Although the coverage is broad, the survey would benefit from presenting some quantitative comparisons (e.g., accuracy, activation memory usage), which would help provide readers with a more concrete understanding of trade-offs across methods. For example, table 1 lists many VP and VPT variants, while VP generally incurs lower computational cost, readers will benefit from guidance on when the extra cost of VPT is justified over cheaper VP, ideally framed in terms of specific regimes or use cases.

**Audience:**

Yes

**Audience Explanation:**

The submission is very well aligned with TMLR's scope.

**Claims And Evidence:**

Yes

**Claims Explanation:**

As a survey paper, most of the claims are drawn directly from the original works.

**Requested Changes:**

Optional, please refer to weaknesses section

---

> ### Author Response · Authors · 2025-12-20
> **Response to Reviewer ajGX**
>
> We sincerely thank the reviewer for the positive feedback, distinguishing our work as a "comprehensive survey" and a "great resource." We are also grateful for the recognition of our taxonomy and the detailed memory analysis in Section 2.4.
>
> Regarding the suggestion to include quantitative comparisons and guidance on the trade-off between VP and VPT, we found this valuable for enhancing the practical utility of our survey. We thus synthesize existing quantitative evidence into actionable guidelines.
>
> **(1) Quantitative Comparisons and Trade-off Guidance**
>
> "The survey would benefit from presenting some quantitative comparisons... readers will benefit from guidance on when the extra cost of VPT is justified over cheaper VP..."
>
> **Response:**
>
> This is a great suggestion. We agree that readers need clear guidance on the "Performance vs. Cost" trade-off. To provide immediate quantitative context, we compiled a comparison of representative methods on four widely-used benchmarks with a ViT-B/16 backbone (see Table R1 below).
>
>
> **Table R1. Performance comparison (Accuracy %) and parameter efficiency of representative VP and VPT methods on 4 widely-used benchmarks with ViT-B/16 backbone.**
>
> | Method | Params (%) | EuroSAT | Flowers | Pets | DTD | Avg. |
> | :--- | :---: | :---: | :---: | :---: | :---: | :---: |
> | **_Visual Prompting (VP)_** | | | | | | |
> | EVP | ~0.77 | 97.6 | 82.3 | 90.0 | 68.4 | 84.6 |
> | AttrVR | ~0.79 | 93.8 | 92.9 | 93.3 | 65.6 | 86.4 |
> | SA2VP | ~0.78 | 95.9 | 99.2 | 92.6 | 75.6 | 90.8 |
> | LoR-VP | ~0.76 | 96.3 | 98.6 | 92.2 | 72.5 | 89.9 |
> | **_Visual Prompt Tuning (VPT)_** | | | | | | |
> | VPT | 0.76 | 96.1 | 98.0 | 88.3 | 65.8 | 87.1 |
> | E2VPT | 0.32 | 96.8 | 98.2 | 88.5 | 67.8 | 87.8 |
> | VFPT | 0.39 | 96.5 | 99.3 | 90.3 | 69.4 | 88.9 |
> | ViaPT | 0.40 | 96.9 | 99.2 | 92.6 | 76.3 | 91.3 |
>
> As shown in Table R1, modern VP methods can match or slightly exceed some VPT baselines. However, VPT variants demonstrate superior parameter efficiency (e.g., VFPT 0.39\% vs. LoR-VP ~0.76\%). This quantitative insight strengthens our new "Guideline for Selection: VP vs. VPT" added to Section 8 (Discussion). Based on findings from key literature [1], we clarify when the extra cost of VPT is justified. While VP is cheaper and black-box friendly, VPT becomes advantageous when adapting to domains with significant distribution shifts, such as medical imaging and remote sensing, where deep parameter recalibration is necessary. From the efficiency perspective, we further discuss scenarios with strict parameter budgets. When the budget is extremely limited ($<$1\%), VPT-deep typically achieves higher accuracy than VP, justifying its additional training memory cost. In addition, we have added Section 2.5, "Comparative Analysis of Generation Mechanisms," which qualitatively compares the performance ceilings and deployment constraints of fixed, learnable, and generated prompts.
>
> We appreciate your thoughtful comments. We hope our response addresses your concerns. Please let us know if there are any additional questions, and we will be happy to discuss further.
>
> *Reference*
>
> [1] "E2VPT: An Effective and Efficient Approach for Visual Prompt Tuning." *ICCV 2023*

---

> > ### Comment · Reviewer_ajGX · 2026-01-02
> >
> > I appreciate the detailed response and do not have any further concerns.

---

> > > ### Author Response · Authors · 2026-01-02
> > > **Reponse to Reviewer ajGX**
> > >
> > > We are delighted to hear that our response has addressed your concerns and received your approval of our work. We sincerely appreciate the effort and time you have dedicated to reviewing our paper, as well as your thoughtful and constructive feedback.

---

### Review · Reviewer_E5A5 · 2025-12-07

**Summary Of Contributions:**

This paper is the first survey dedicated specifically to prompt-based adaptation (PA) for large vision models. It also establishes a well-organized taxonomy for PA methods, based on prompt-generation mechanisms (fixed, learnable, generated), various CV tasks, specific domains, and constrained learning scenarios.

# Strengths
- The paper effectively discusses and compares the differences with prior surveys, outlining how its focus and contributions differ from existing work.
- This paper provides well-structured tables and informative visual diagrams to summarize and organize the different PA methods.
- The writing is clear and easy to understand for the reader

# Weaknesses
- It is questionable to treat injection granularity (pixel-level vs. token-level) as an independent categorization dimension. As shown in Table 1, all pixel-level approaches fall under VP and all token-level approaches fall under VPT, making this distinction essentially overlapping with the VP/VPT division rather than offering a meaningful axis of classification.
- The paper’s analytical depth is somewhat limited. Except for the efficiency discussion in Section 2.4, the paper lacks a fuller comparison of different types of methods (VP-fixed, VP-learnable, and VP-generated) across other important dimensions, such as performance trade-offs, inherent limitations, or practical deployment constraints.
- The taxonomy introduced in Section 2 (fixed vs. learnable vs. generated prompts) is not consistently connected to later sections. Section 3, for example, discusses applications without relating them back to these classifications, reducing cross-sectional coherence.

**Audience:**

Yes

**Audience Explanation:**

This paper offers a comprehensive and well-organized summarization of existing work on prompt-based adaptation for vision models, addressing the missing gap in the current literature. It gives readers a clear, high-level understanding of the field and presents an informative framework for navigating research in this area.

**Claims And Evidence:**

Yes

**Claims Explanation:**

This paper provides a structured summarization of existing prompt-based adaptation methods for vision models. Authors include a broad and diverse set of studies to build a comprehensive and coherent taxonomy of the field.

**Requested Changes:**

- **Revise the abstract’s description of “injection granularity.”** The current phrasing suggests that pixel-level vs. token-level injection is an additional, informative categorization axis. However, as reflected in Table 1, all pixel-level methods correspond to VP and all token-level methods correspond to VPT, making this dimension effectively redundant with the VP/VPT distinction. The abstract should adjust its wording to avoid implying that injection granularity is an independent or orthogonal classification criterion.
- **Add a dedicated paragraph offering a clearer comparison among summarized methods (fixed, learnable, and generated).** Beyond the efficiency discussion in Section 2.4, the paper would benefit from a more thorough analysis of the performance implications, limitations, and practical constraints associated with each category.
- **Strengthen the connections between the taxonomy in Section 2 and the application discussions in later sections.** The fixed/learnable/generated classification is not referenced in Section 3 and beyond, resulting in weak cross-sectional coherence. Explicitly linking the taxonomy to specific applications would enhance clarity and structural consistency.

---

> ### Author Response · Authors · 2025-12-20
> **Response to Reviewer E5A5**
>
> We sincerely thank the reviewer for the thoughtful feedback, particularly for appreciating our well-organized taxonomy and informative visual diagrams. We have revised the manuscript to address the concerns as follows.
>
> **(1) Revise Abstract’s description of injection granularity**
>
> "The abstract should adjust its wording to avoid implying that injection granularity is an independent or orthogonal classification criterion."
>
> **Response:**
>
> Thank you for pointing it out. We agree that treating *injection granularity* as an independent classification axis creates redundancy, as it shares a direct relation with the VP/VPT distinction in our taxonomy. We have rewritten the Abstract to explicitly clarify that VP corresponds to pixel-level injection, while VPT operates at the token level.
>
> **(2) Clearer comparison among summarized methods (Fixed, Learnable, and Generated)**
>
> "The paper would benefit from a more thorough analysis of the performance implications, limitations, and practical constraints associated with each category."
>
> **Response:**
>
> To provide the requested analytical depth, we have added Section 2.5, *Comparative Analysis of Generation Mechanisms,*
> immediately following the efficiency discussion. This section analyzes the trade-off in adaptability, deployment, and performance. Generally speaking, fixed prompts are effective in black-box and interactive scenarios, but are quite limited under systematic domain shifts. Learnable prompts enable domain adaptation via gradient-based optimization but remain static and suboptimal for instance-level variation. Generated prompts offer superior instance-awareness and generalization in dynamic environments at the cost of increased inference latency and optimization complexity.
>
> **(3) Strengthen connections between taxonomy and applications**
>
> "Explicitly linking the taxonomy to specific applications would enhance clarity and structural consistency."
>
> **Response:**
>
> This is a great suggestion. We have revised Section 4 (Applications) to explicitly map applied methods back to the taxonomy defined in Section 2. Each domain subsection now concludes with a *Summary and Insights* paragraph linking applications to Fixed, Learnable, and Generated prompt categories. For example, in Medical Imaging, fixed VP dominates interactive segmentation, while generated VPT is increasingly used for patient-specific variation.
>
> We appreciate your thoughtful comments. We hope our response addresses your concerns. Please let us know if there are any additional questions, and we will be happy to discuss further.

---

> > ### Comment · Reviewer_E5A5 · 2026-01-01
> >
> > I appreciate the authors' response! I have no further concerns.

---

> > > ### Author Response · Authors · 2026-01-01
> > > **Reponse to Reviewer E5A5**
> > >
> > > We are delighted to hear that our response has addressed your concerns and received your approval of our work. We sincerely appreciate the effort and time you have dedicated to reviewing our paper, as well as your thoughtful and constructive feedback.

---

### Review · Reviewer_G9N7 · 2025-12-15

**Summary Of Contributions:**

This paper presents a comprehensive survey of Prompt-based Adaptation(PA) for Large scale vision models. The main contribution is a unified and structured taxonomy that distinguish between visual prompting (VP) and Visual Prompt Tuning (VPT), where VP operates in the input space, VPT injects learnable prompt tokens into the internal representations of the backbone.
The paper reviews PA applications across a broad range of challenging domains, including medical imaging, robotics and embodied AI, remote sensing, and autonomous driving. Furthermore, the text addresses how PA contributes to Trustworthy AI, specifically regarding robustness, fairness, and security. Finally, the paper list current challenges such as training instability and inference latency.

**Audience:**

Yes

**Audience Explanation:**

Yes. This survey organizes the literature from a novel and well-motivated perspective, and I found it informative and helpful to read. Given the growing interest in parameter-efficient adaptation of large scale vision models, it is likely that many readers in the TMLR audience would benefit from the unified taxonomy and the clear overview provided in this paper.

**Broader Impact Concerns:**

N/A. This survey primarily synthesizes existing work and does not introduce new technical capabilities, so it does not raise significant new ethical concerns beyond those already present in the underlying literature.

**Claims And Evidence:**

Yes

**Claims Explanation:**

The paper's main claims are generally well supported. (1) The authors build the taxonomy with accurate references and comprehensive coverage of prior work; (2) the detailed literature review, organized by domain, effectively supports the discussion on cross-domain effectiveness; (3) the arguments regarding efficiency and practical constraints are validated through comparisons of parameter usage and adaptation methods; and (4) the theoretical foundations and open challenges are effectively substantiated by synthesizing empirical evidence on model behavior and robustness.

**Requested Changes:**

The submission is a high-quality survey with comprehensive coverage and solid analysis. To further polish the work and maximize its utility for the community, I suggest the following adjustments (these are to strengthen the work):

(1) Strengthen Taxonomy Alignment: The discussion in the "Applications" section currently reads somewhat independently from the "Unified Taxonomy" proposed earlier. It would be better if the authors explicitly categorize the applied methods in these domains using their own VP/VPT definitions. Specifically, for each domain (e.g., Medical, Robotics), clarify whether pixel-level (VP) or token-level (VPT) approaches are dominant and explain why that specific methodology fits the domain's constraints;

(2) Also, the application section seems longer compared to the methodological sections and tends to be list-like.

(3) Instead of merely reporting that PA is used, the authors should focus on comparative analysis, such as discussing which type of prompting (VP vs. VPT) yields better performance or efficiency trade-offs in specific downstream scenarios to reconnect with the paper's core evaluation.

---

> ### Author Response · Authors · 2025-12-20
> **Reponse to Reviewer G9N7**
>
> We sincerely thank the reviewer for the positive review, recognizing it as a "high-quality" survey with "comprehensive coverage" and "solid analysis". We are particularly grateful for your constructive suggestions regarding the structure and depth of the application section. These insights have helped us significantly strengthen the connection between our taxonomy and the practical deployment of PA. We provide explanations to your questions point-by-point in the following.
>
> **(1) Strengthen Taxonomy Alignment**
>
> "Explicitly categorize the applied methods in these domains using their own VP/VPT definitions... clarify whether pixel-level (VP) or token-level (VPT) approaches are dominant and explain why..."
>
> **Response:**
>
> Thank you for your valuable suggestion.
> To better bridge the gap between the proposed unified taxonomy and the application review, we have extensively revised Section 4 (Applications). Specifically, we have added a dedicated *Summary and Insights* paragraph at the end of each domain subsection (e.g., Medical, Remote Sensing, Robotics) to showcase the underlying reasons for these categorizations. For instance, we discuss that VP dominates in interactive medical segmentation due to the need for spatial cues, while VPT is preferred in remote sensing for bridging massive domain gaps via deep semantic realignment. This ensures the methodological taxonomy is consistently applied throughout the application discussion.
>
> **(2) Application Section Structure (List-like \& Length)**
>
> "The application section seems longer compared to the methodological sections and tends to be list-like."
>
> **Response:**
>
> Thank you for pointing it out. We would like to politely clarify that the length of the application section stems from our goal to ensure comprehensiveness of PA-based attempts, given the explosive growth of PA-related literature across diverse fields. However, we agree that a simple enumeration reduces the survey's readability. To address this, we have restructured the narrative to group related works by shared technical challenges rather than listing them sequentially. We have synthesized redundant descriptions and merged citations where appropriate, making the section more compact and narrative-driven while retaining necessary coverage.
>
> **(3) Comparative Analysis (VP vs.\ VPT Trade-off)**
>
> "Focus on comparative analysis... discussing which type of prompting yields better performance or efficiency trade-off..."
>
> **Response:**
>
> This is a great question. To move beyond descriptive reporting, we have added a new subsection, *Guideline for Selection: VP vs.\ VPT,* at the beginning of Section 8 (Discussion). This subsection provides a structured comparative analysis, outlining specific scenarios (e.g., black-box access, deep semantic adaptation, efficiency constraints) where one approach clearly outperforms the other.
>
> We appreciate your thoughtful comments. We hope our response addresses your concerns. Please let us know if there are any additional questions, and we will be happy to discuss further.

---

> > ### Comment · Reviewer_G9N7 · 2025-12-20
> >
> > I appreciate the authors quick response, my concern has been cleared.

---

> > > ### Author Response · Authors · 2025-12-21
> > > **Reponse to Reviewer G9N7**
> > >
> > > We are delighted to hear that our response has addressed your concerns and received your approval of our work. We sincerely appreciate the effort and time you have dedicated to reviewing our paper, as well as your thoughtful and constructive feedback.

---

### Decision · Action_Editor_9Gwx · 2026-01-16

**Recommendation:** Accept as is

**Audience:**

Yes

**Audience Explanation:**

This paper studies the prompt-based adaptation which is interesting and important for LLM engineering. The survey covers a broad enough spectrum and is of the interests of TMLR audience.

**Claims And Evidence:**

Yes

**Claims Explanation:**

This is a survey paper with solid literature covered and good theoretical foundations. The claims are therefore well supported.